# Semi-volatile and highly oxygenated gaseous and particulate organic compounds observed above a boreal forest canopy

Ben H. Lee[1], Felipe D. Lopez-Hilfiker[1,†], Emma L. D'Ambro[2], Putian Zhou[3], Michael Boy[3], Tuukka Petäjä[3], Liqing Hao[4], Annele Virtanen[4], Joel A. Thornton[1]

[1]Department of Atmopsheric Sciences, University of Washington, Seattle, WA, U.S.A.
[2]Department of Chemistry, University of Washington, Seattle, WA, U.S.A.
[3]Institute for Atmospheric and Earth System Research / Physics, Faculty of Science, University of Helsinki, Finland
[4]Department of Physics, University of Eastern Finland, Kuopio, Finland
[†]Now at TofWerk AG, Thun, Switzerland

*Correspondence to*: Joel A. Thornton (thornton@atmos.uw.com)

**Abstract.** We present hourly on-line observations of molecular compositions ($C_xH_yO_zN_{0-1}$) and abundances of oxygenated organic species in gas and submicron particle phases from April to June of 2014 as part of the Biogenic Aerosols-Effects on Cloud and Climate (BAECC) campaign. Measurements were made using the Filter Inlet for Gases and AEROsols coupled to

a high-resolution time-of-flight iodide-adduct ionization mass spectrometer (FIGAERO-CIMS) located atop a 35 m tall tower, about 10 m above a boreal forest canopy at the SMEAR II research station in Hyytiälä, Finland. Semi-volatile and highly oxygenated multifunctional (HOM) organic species possessing from 1 up to 20 carbon atoms, and with as few as 2 and as many as 16 oxygen atoms were routinely observed. Utilizing non-negative matrix factorization, we determined that >90% and >99% of the organic mass in the gas and particle phases, respectively, exhibited one of three distinct diel trends;

one in which abundances were enhanced at daytime, another in the early morning hours, and thirdly during nighttime. Particulate organic nitrates contributed ~35% to the total organic aerosol mass loading at night during BAECC, much higher than observed by the same instrument package at a mixed-deciduous forest site in the Southeast U.S that experienced higher nighttime concentrations of nitrogen oxides. Unique HOM monomers (defined here as those with 10 carbon and 7 or more oxygen atoms) and dimers (at least 16 carbon atoms), with and without a nitrogen atom, were found in most of the three

subgroups of both phases. We show the potential to connect these groupings of compounds based on their distinct behavior in time to the expected chemical conditions (biogenic VOC precursor, oxidant type, etc.) responsible for their production. A suite of nitrated dimer-like compounds was detected in both the gas and particle phases, suggesting a potential role for the formation of low volatility organics from $NO_3$ radical driven, as well as daytime NO-influenced, monoterpene chemistry.

# 1 Introduction

The world's forests emit about a petagram of C per year in the form of hydrocarbons ($C_xH_y$) [*Guenther et al.*, 2006], an amount comparable to that stored annually in the biome due to the growth of organic matter [*Steffen et al.*, 1998]. Isoprene ($C_5H_8$) and monoterpenes ($C_{10}H_{16}$) together account for more than half of the total biogenic hydrocarbon emissions [*Guenther et al.*, 2012]. Upon their release to the atmosphere, they undergo oxidation reactions over timescales of several minutes to hours during which hundreds, if not thousands, of products possessing unique molecular compositions ($C_xH_yO_zN_{0,1}$) are generated due to fragmentation, functionalization and/or accretion [*Goldstein and Galbally*, 2007; *Kroll et al.*, 2011]. Identifying the molecular compositions of the cascade of compounds unleashed during this process is key to determining their chemical properties such as saturation vapor pressure and reactivity, which are fundamental to assessing their potential for forming and growing nanometer sized atmospheric particles and the timescales of their influence downwind of the region of emission.

Boreal forests, located in the mid to high latitudes, are undergoing rapid warming [*Bonan*, 2008] with longer growing seasons that act to strengthen the emission rates of biogenic volatile organic compounds (VOCs) [*Guenther et al.*, 2006]. It is widely recognized that the condensation or reactive uptake of low-volatility organic vapors, derived from the oxidation of BVOC, drives the growth of Aitken mode particles to cloud condensation nuclei (CCN) sizes in remote continental and some coastal regions near marine stratus [*Hallquist et al.*, 2009; *Jimenez et al.*, 2009; *Riipinen et al.*, 2011]. A thorough understanding of aerosol particle formation rate, particle growth rate, lifetime and fate is, therefore, crucial for quantitatively assessing their impact on CCN activity that in turn influences regional radiative and hydrological budgets [*Heald et al.*, 2008; *Paasonen et al.*, 2013; *Spracklen and Rap*, 2013]. Large regions of the world's boreal forests, relative to temperate forests, are (i) minimally affected by anthropogenic pollutants, which can alter the chemical pathways by which biogenic VOC oxidation proceeds, and (ii) emit large quantities of terpenes, namely monoterpenes and sesquiterpenes ($C_{15}H_{24}$), that, once oxidized, readily form condensable material that leads to the formation and growth of secondary organic aerosol (SOA) [*Ehn et al.*, 2014; *Kavouras et al.*, 1998].

We deployed the Filter Inlet for Gases and AEROsols (FIGAERO) coupled to a high-resolution chemical ionization time-of-flight mass spectrometer (HRToF-CIMS) as part of the Biogenic Aerosols Effects on Clouds and Climate (BAECC) campaign, which took place in Hyytiälä, Finland during April and May of 2014. The FIGAERO HRToF-CIMS, henceforth referred to as FIGAERO-CIMS, provided on-line measurements of molecular compositions ($C_xH_yO_zN_{0-1}$) and abundances of the organic constituents of the gas and particle phases. We organize all of the organic constituents of the gas and particle phases into subgroups that are characterized by their behavior in time by utilizing a matrix factorization approach, similar to that of *Yan et al.* [2016] which employed positive matrix factorization (PMF) on gas-phase measurements made using the Atmospheric Pressure interface Time-Of-Flight mass spectrometer (APi-TOF, Aerodyne Research Inc. & Tofwerk AG;

[*Junninen et al.*, 2010]). A number of previous studies have presented observations of the organic constituents of the gas-phase at this site [*Bianchi et al.*, 2017; *Ehn et al.*, 2010; *Ehn et al.*, 2012]. The hourly measurements of both phases afforded by the FIGAERO-CIMS allow us to more accurately determine the temporal behavior of organic molecules in the particle-phase, and the relationship to their gas-phase counterparts.

**2 Methods**

Observations were made over a boreal forest in Hyytiälä, Finland (61° 50' 36.73" N, 24° 17' 16.25" E) at the Station for Measuring Ecosystem-Atmosphere Relations (SMEAR II), a long-term field site dedicated to comprehensive measurements of reactive gases and aerosol particle characteristics since its inception in 1996 [*Hari and Kulmala*, 2005; *Kulmala et al.*, 2000; *Junninen et al.*, 2009]. BAECC was an intensive field campaign organized by the U.S. Department of Energy and the

University of Helsinki, and took place from February to September of 2014 with the primary goal of assessing the sources and effects of aerosol particles formed from biogenic VOC [*Petäjä et al.*, 2016]. The history, stand age, tree species composition and other aspects of the forest enveloping the SMEAR II site have been detailed in previous studies [*Ilvesniemi and Liu*, 2001; *Kulmala et al.*, 2001]. The deployment period for the FIGAERO-CIMS was between mid-April to early June of 2014, and will be the focus here.

The HRToF-CIMS instrumentation [*Junninen et al.*, 2010; *Lee et al.*, 2014; *Yatavelli et al.*, 2012], the FIGAERO front end [*Lopez-Hilfiker et al.*, 2014], their operation in the field coupled together [*D'Ambro et al.*, 2017; *Liu et al.*, 2016; *Lopez-Hilfiker et al.*, 2016b], as well as its deployment during BAECC [*Mohr et al.*, 2017; *Schobesberger et al.*, 2016] are detailed elsewhere. Briefly, the FIGAERO-CIMS was located at the top of a 35-m tall scaffolding tower, on the southwestern edge of

the top platform. Ambient particles were drawn in through a 2 m long stainless steel inlet (22 mm inner diameter) at a flow-rate of 2.5 slpm. A custom inertial impactor was used to remove particles > ~2 μm before collection on a 1-μm pore size perfluorotetrafluoroethylene (PTFE) filter in the FIGAERO unit. Particle collections were conducted for 30 minutes, during which the gas-phase was measured by sampling ambient air at a flow-rate of 22 slpm through a 1 m long PTFE inlet (17 mm inner diameter). Particles were desorbed off of the FIGAERO filter by heating ultrahigh-purity $N_2$ (2.5 slpm through the

FIGAERO PTFE filter) by 10°C min$^{-1}$ up to 200°C. The desorption cycle lasted 60 minutes, during which ~20 slpm was maintained to avoid stagnant air in the gas-phase inlet. Every fourth particle collection cycle was conducted with an additional particle filter upstream of the normal FIGAERO filter. This provided a way to correct for interfering background signals arising from semi- and nonvolatile gases that can collect on filters and from the ionization source. A high-resolution time-of-flight Aerosol Mass Spectrometer (AMS) [*DeCarlo et al.*, 2006; *Dunlea et al.*, 2009] was located inside a ground-

based trailer near the base of the scaffolding tower. The copper inlet (4.4 mm inner diameter) to the AMS instrument was located 3.7 m off the canopy floor. The total flow through the AMS inlet was 1.09 slpm.

Given the challenges associated with obtaining reliable calibration sources for each of the hundreds of unique molecular compositions present upon oxidation of VOCs [*Goldstein and Galbally*, 2007; *Hunter et al.*, 2017] and detected by iodide-adduct ionization [*Isaacman-VanWertz et al.*, 2017; *Lopez-Hilfiker et al.*, 2016a], we do not attempt to close any mass budgets with the FIGAERO-CIMS measurements alone. For the purposes here, we apply a maximum sensitivity the instrument is capable of measuring any particular compound in order to obtain a lower limit on the concentration that is accounted for by the detected ion intensity within the mass spectrometer. The maximum sensitivity is determined by the highest rate of collision between the iodide reagent ion and the compound of interest in the ambient matrix, *i.e.*, assuming formation of the adduct ion at the collision limited rate, no fragmentation, surface reactions or losses of charge once clustered with the iodide ion [*Lopez-Hilfiker et al.*, 2016a]. While the collision cross section and thus the collision frequency will vary from compound to compound, we neglect this effect as it is of order of factor of 2, and instead quote an uncertainty in the minimum concentration of ±50%. The focus of the work here is the distributions of molecular compositions that comprise the gas and particle phases with a specific emphasis on highly oxygenated molecules (HOM), how they evolve in time distinctly from one another and the interpretation of these observations for their sources and sinks.

For the 584 identified organic carbon species (OC=$C_xH_yO_zN_0$), $x$ ranges from 1 to 20, $y$ is an even number greater than or equal to $x$ but less than or equal to $2x+2$, and $z$ is greater than or equal to 2. For the 434 identified organic nitrate species (ON=$C_xH_yO_zN_1$), $x$ also ranges from 1 to 20, $y$ is an odd number greater than or equal to $x$ but less than $2x+2$, and $z$ is greater than or equal to 4. We note that there are more ion peaks than represented here at the higher mass-to-charge ratios where the carbon number exceeds 20, but the resolution of the mass spectrometer at those ranges is not sufficient to allow confident composition assignment. The fraction of mass concentration in the particle-phase that is unassigned is < 10%.

We utilized non-negative matrix factorization (NNMF) built into the MATLAB computing software [*Berry et al.*, 2007] to determine what the general diurnal behaviors were during BAECC and to which each of the 1018 organic species comprising the gas and particle phases belonged. NNMF is analogous to Positive Matrix Factorization (PMF), in that it explicitly describes the variability of the input data matrix with a reduced number of factors [Paatero and Tapper, 1994] to yield non-negative solutions. *Yan et al.* [2016] demonstrated the utility of the PMF technique on measurements of ELVOC measured by APiTOF CIMS [*Junninen et al.*, 2010]. The goals of the NNMF technique are similar, that is, to determine the type and number of groups that behave uniquely in time. One distinction is that the uncertainties associated with each of the input observations are currently not utilized by the NNMF appraoch as opposed to PMF. NNMF was performed separately for the organic species in the gas-phase from those in the particle-phase.

First, the hourly median of the deviation from the daily mean for each of the 1018 organic species in the gas-phase and the particle-phase were determined, which effectively imparts equal weight to all species. Then for each phase, NNMF was implemented on the 24 ($m$) × 1,081 ($n$) matrix with up to 23 ($n$–1) factors yielding 23 diel trends (W) and their

corresponding weights (H), or the degree to which each of the 1,081 vectors belonged to the 23 trends. Not all 23 trends were statistically unique from one another or represented real atmospheric behavior, given that NNMF attempts to explicitly solve for the variability of the input matrix with $n$–1 set of products of the trends (W) and weights (H). (More weight is given by NNMF to input vectors of greater magnitude, but since the input matrix here is the deviation from the daily mean, each species is given more or less equal weight.) That is to say, NNMF does not distinguish between signal and artifact or noise. To do so, we incrementally lowered the factor number ($n$–1) of each NNMF computation until a satisfactory set of diel trends that were unique in terms of atmospheric behavior were determined. Each species was identified as belonging to not more than one trend, henceforth called subgroup, and exhibited correlation coefficient ($R^2$) with that subgroup greater than 0.45. Species that did not meet this criterion were designated into an "others" subgroup. The robustness of this approach is easily verified by visualizing the diel trends of the subset of the input distinguished by these identifiers. The species in the "others" subgroup typically exhibited little to weak diel trends, likely affected by noise due to low signal.

A result of such a conservative approach is that species exhibiting subtle differences in temporal trends may be lumped into a single subgroup. Our goal with the implementation of NNMF was to identify broadly distinct trends that explain the behaviors of the majority of organic species detected by the iodide-ionization method without pre-grouping based upon their molecular composition (carbon atom number, oxygen to carbon ratio, etc.). We do, however, distinguish between the organic carbon and organic nitrate groups as they are products of distinctly different oxidation schemes.

## 3 Results

### 3.1 Overview of detected compounds

A total of 1018 compounds possessing unique molecular compositions were identified during BAECC. The mixing ratio distribution in the gas-phase generally decreased with increasing molecular weight, whereas in the particle-phase, compounds of higher molecular weights comprised a greater fraction of the total mass (figure 1), consistent with the idea that species of higher molecular weight more readily condense than those of lower molecular weight. The effective, or mixing ratio-weighted, molecular weight of the gas-phase was 144 g mol$^{-1}$, whereas the effective molecular weight of the particle-phase was 221 g mol$^{-1}$. A number of relatively low molecular weight species (< 125 g mol$^{-1}$) were observed at levels greater than expected in the particle-phase compared to those of higher molecular weights that, as a collective, typically exhibits an approximate bell-shape distribution in abundance (figure 1). Those 50 species – accounting for on median <7% of the total particulate organic mass as measured by the FIGAERO-CIMS – possibly originated from fragmentation of larger molecular weight compounds during thermal desorption [*Lopez-Hilfiker et al.*, 2014]. In the event that fragmentation of a large molecule during thermal desorption yielded multiple fragments that were not all detected by the FIGAERO-CIMS, then this may explain why observations by the FIGAERO-CIMS was at minimum about half of the organic aerosol mass measured by

an Aerosol Mass Spectrometer [*Lopez-Hilfiker et al.*, 2016b]. The goal of this analysis given the suite of organic constituents measured is to determine their unique diel trends, which are unlikely to be driven by the effects of thermal fragmentation.

The FIGAERO-CIMS utilizing iodide-adduct ionization detected HOM monomers, defined here as those possessing 7 or more oxygen atoms, as well as dimers possessing at least 16 carbon atoms, with and without a nitrogen atom, in both the gas and particle phases (insets of figures 1a and 1b). The diel and day-to-day variability of their ambient concentrations were related to those of the ambient air temperature (figure 2), due likely to their being byproducts of monoterpene oxidation (since BVOC emission rates are dependent on ambient temperature) but also in part to the temperature dependence of the rate of autoxidation [*Crounse et al.*, 2011], which presumably increases the yields of HOM species. The mixing ratios of HOM monomers and dimers observed above the forest canopy with iodide-ionization were comparable to those observed at the same site in May to November of 2010 using the APi-TOF below the canopy [*Ehn et al.*, 2012].

Approximately 90% of the detected gas-phase mass was comprised of organic compounds possessing 10 or less carbon atoms (insets in figures 3a and 3b). The abundance of species with 10 carbon atoms that comprised gaseous organic nitrates (gON=$C_xH_yO_zN_1$) and organic carbon (gOC=$C_xH_yO_zN_0$) decreased with increasing molecular weight (figures 3a and 3b), consistent with the general trend observed for all species in the gas-phase (figure 1a). The decrease in abundance with increasing oxygen atom number was largely independent of carbon atom number, as summarized for the $C_{1-7}$, $C_{8-10}$ and $C_{11-20}$ groups shown in the insets of figures 3a and 3b. This trend likely reflects the effects of decreasing volatility with additional attachment of an oxygen-containing functional group or accretion, and suggests that the yield of a given multifunctional organic compound from its presumed less-oxygenated precursor of the same carbon atom number is less than 0.5. Otherwise, the abundance distribution of species of a given nC would increase with every addition of an oxygen atom number (nO).

The detected particle-phase mass was mostly (~82%) in compounds possessing 10 or less carbon atoms (insets of figures 3c and 3d). In contrast to the gas-phase, however, the mass contribution from compounds with higher oxygen atom number was greater with increasing carbon atom number (insets of figures 3c and 3d). This effect was more pronounced for particulate organic carbon (pOC=$C_xH_yO_zN_0$) than it was for particulate organic nitrates (pON=$C_xH_yO_zN_1$). Species that comprised pON and pOC within a given nC group generally exhibited an approximate bell-shape distribution as a function of molecular weight, with those possessing 5 to 8 oxygen atoms representing the apex within each nC group (figures 3c and 3d), consistent with observations from a mixed-deciduous forest in the Southeast U.S. [*Lee et al.*, 2016]. The bell-shaped distribution of oxygen number for a given nC group observed in the particle phase is consistent with the decaying abundance of gas-phase species with oxygen number and the increasing thermodynamic driving force for such compounds to partition to the condensed phase. This notion assumes that the oxygen addition to hydrocarbon precursors occurs in the gas-phase. Gaseous and particulate organic carbon compounds (gOC and pOC) with 11 to 20 carbon atoms (insets of figures 3b and 3d) exhibited two modes with respect to oxygen atom number, where there was a noticeable decrease in abundance of those

possessing 7 oxygen atoms compared to those with 4-6 and 8-10. This may have been due to the combined effects of auto-oxidation leading to additions of $O_2$ following OH or $O_3$ initiated oxidation of the parent BVOC and $RO_2$-$RO_2$ reactions that did not favor the formation of $C_{11-20}$ dimer compounds with 7 oxygen atoms.

### 3.2 Diel trends: gas-phase

The complex and congested array of products present in both phases (figure 1) motivates the use of factorization techniques to reduce observed spectra into a smaller set of co-varying components having similar attributes. We present results from NNMF analyses that categorize each of the gON, gOC, pON and pOC groups into subgroups defined by their unique behavior in time. Resulting diel patterns and day-to-day variations in the relative importance of different subgroups can be connected to expected shifts in precursor emissions, oxidant type ($O_3$, OH, $NO_3$, etc.), peroxy radical fate (reaction with

$HO_2$, $RO_2$, NO, or isomerization), and meteorological conditions (ambient temperature, boundary layer height, extent of mixing between above and below forest canopy, etc.) that affect gas-particle partitioning and multiphase chemistry.

In the gas-phase, a total of 714 out of the 1018 identified species belonged to one of the three subgroups characterized by their unique diel trends (figure 4), determined as described in the methods section. One subgroup exhibited a diel trend in

which the ambient levels were enhanced at midday, another in which the levels were enhanced in the morning hours, and another during nighttime, henceforth, referred to as the daytime, morning and nighttime subgroups. The names of these groups should not be interpreted as definitive assignment to the timing of their production, but rather the timing of their collective enhancements at the measurement location. For example, a set of compounds produced overnight in the nocturnal residual layer may not be observed at the top of the instrument tower until mid-morning at the break-up of the nocturnal

boundary layer.

The daytime, morning and nighttime subgroups, which comprised of 602, 92 and 20 organic species, respectively, accounted for a median 78%, 8.7% and 3.0%, respectively, of the total measured gas-phase mixing ratio. The daytime subgroup exhibited the lowest effective molecular weight relative to the other two subgroups (table 1), due possibly to OH oxidation in

the presence of NO (figure S1) that favors formation of alkoxy radicals and subsequently C-C bond scission products over those yielding HOM species [*Seinfeld and Pandis*, 2016]. The daytime diel trend is opposite that typically exhibited by monoterpene mixing ratio [*Hakola et al.*, 2012], but consistent with that of their emission rate and levels of oxidant such as ozone ($O_3$) and the hydroxyl radical (OH) [*Spanke et al.*, 2001]. *Holzinger et al.* [2005] also observed levels of BVOC oxidation products enhanced at daytime above the canopy at a monoterpene-emitting pine forest in California. Such trends

imply that formation rates of these oxidation products were sufficiently higher during the day than night in order to overcome the lower parent BVOC concentrations and greater boundary layer height during the day, consistent with modeling results that were specific to the SMEAR II site [*Smolander et al.*, 2013].

Semi-volatile and HOM organic species enhanced during the morning period were likely produced by chemistry favored under conditions of lower $RO_2$:NO ratios than those at daytime, considering BVOC emission rates were weaker and NO levels were elevated due to photolysis of $NO_x$ that had accumulated in the nocturnal surface layer [*Horii et al.*, 2004; *Min et*
*al.*, 2014] (figure S1). As such, the effective nitrogen atom number (nN) of the morning gas-phase subgroup was 0.5, that is, half of the number density was comprised of organic nitrates, a value higher than those exhibited by the day and night subgroups (table 1). Entrainment of intra-canopy air and/or air above the nocturnal surface layer, not sampled at night due to slow mixing, may have also contributed to the morning subgroup.

Only 20 gaseous species (figure 4e), none classified as a HOM species, were elevated at night. Out of those 20, 9 were ON species that comprised on median about 20% of the nighttime subgroup (effective nN=0.2, see table 1), even though levels of $NO_x$ and monoterpenes are typically higher at night when $NO_3$ initiated chemistry occurs. In the particle-phase, however, 125 species, including many HOM exhibited nighttime enhancements (figure 5e). Out of those 125, 97 were organic nitrates (table 1). The fact that species expressing one diel trend in the gas-phase did not strictly follow the same diel trend in the
particle-phase suggests a process more complex than equilibrium-driven gas-particle partitioning for some compounds, possibly multiphase chemistry. The remaining 360 out of the total 1018 species in the gas-phase did not exhibit strong enough diel trends to qualify being categorized in any of the three subgroups due in part to their low abundance relative to instrument detection limit. These remaining 360 species accounted on median 9.8% of the total gas-phase mixing ratio.

The diel trends of the three gas-phase subgroups determined here are similar to those of the three factors obtained using Positive Matrix Factorization by *Yan et al.* [2016], which report on measurements made at the same research site during spring/summer of 2012. The abundances accounted for by each subgroup, however, differ. The iodide-ionization technique above the forest canopy deployed during spring/summer of 2014 (and reported here) observed most gaseous species to belong to the daytime subgroup (figure 6c), whereas measurements near the canopy floor by nitrate-ionization during
spring/summer of 2012 observed nearly equal distribution amongst the three diel-sorted factors [*Yan et al.*, 2016]. And while molecules designated by *Yan et al.* [2016] as daytime 'fingerprint' molecules – such as $C_{10}H_{15}NO_{8,11}$, $C_3H_5NO_6$, $C_4H_5NO_7$, $C_5H_7NO_7$, $C_6H_9NO_7$, and $C_7H_9NO_7$ – were also observed above the forest canopy to exhibit daytime diel trends, two compounds designated by *Yan et al.* [2016] as nighttime 'fingerprint' molecules – $C_{10}H_{14}O_7$ and $C_{10}H_{14}O_9$ – were observed above the forest canopy to exhibit daytime diel trends. The reason for the differing trends is unclear without conducting a
side-by-side inter-comparison. But it is possible that iodide- and nitrate-ionization methods detect distinct isomers that are governed by different chemistry, or that the vertical gradients, particularly at night, confound comparison between measurements made above and below the forest canopy [*Zha et al.*, 2018; *Schobesberger et al.*, 2016].

The contributions of each of the subgroups to the total gas-phase mixing ratio exhibit distinct diel trends (figure 6), clearly reflecting the evolution on the timescale of hours of the chemical processes that govern air mass composition. The gOC group was dominated by the daytime subgroup (figure 6c), whereas the gON group experienced a greater contribution from the morning and nighttime subgroups (figure 6a). The contribution of each subgroup to the total, outside of the time period of its maximum enhancement, does not go to zero (figure 6). This suggests either the chemistry responsible for a given subgroup continues but is slower throughout the rest of the day and/or the products possess lifetimes that are sufficiently long that they are still present after the chemistry responsible for their formation has diminished in a relative sense.

### 3.3 Diel trends: particle-phase

A total of 976 out of the 1018 identified species that were detected in the particle-phase belonged to one of the three subgroups as characterized by their diel trends (figure 5). The daytime, morning and nighttime subgroups were comprised of 519, 332 and 125 organic compounds, respectively, and accounted for a median 51%, 43% and 5.3%, respectively, of the total particle-phase mass concentration (table 1). The relative abundance for each of the three subgroups of the particle-phase exhibited a bell-shaped distribution, while that of the three subgroups of the gas-phase generally decreased with molecular weight (figure 7), as it was similarly observed for $C_{10}$ species (figure 3) and for all species as a collection (figure 1). The nighttime subgroup of the particle-phase possessed the highest effective molecular weight, followed by the morning and daytime subgroups (figure 7). The nighttime subgroup of the particle-phase exhibited the greatest effective nN (=0.9), that is, 90% of the particle mass was constituted by organic nitrates, whereas it was the morning subgroup of the gas-phase that exhibited the greatest nN (=0.5), as shown in table 1. Additionally, pON was on median comprised roughly in equal parts by the three subgroups, whereas gON was dominated by the daytime subgroup with some contribution from the morning subgroup (figure 6b). A similar set of disparities in the contributions from the various subgroups of gOC (figure 6c) versus those of pOC (figure 6d) was also apparent. These observations, as noted above, also highlight the complexity of the relationship that likely exists between the organic constituents of the gas and particle phases.

Total organic aerosol and total nitrate mass loadings were measured from below the forest canopy using an Aerosol Mass Spectrometer [*DeCarlo et al.*, 2006; *Dunlea et al.*, 2009]. There was good agreement (slope=1.09, $R^2$=0.37) in pON measured by the two techniques, assuming that all of the particle-phase nitrate mass measured by the AMS was due to organic nitrates, and applying an average molecular weight of 265 g mol$^{-1}$ (effective molecular weight of pON as measured by FIGAERO-CIMS) to the AMS nitrates (figure 8b). The pON measured by the two instruments exhibited similar diel trends, with the maxima reaching in the early morning hours near sunrise. For total organic aerosol, the FIGAERO-CIMS detected approximately half of that observed by the AMS (slope=0.66, $R^2$=0.23), similar to previous comparisons between the two techniques [*Lopez-Hilfiker et al.*, 2016b] (figure S2). The diel trends exhibited by AMS organics and FIGAERO-CIMS pON+pOC, however, showed markedly different diel trends (figure S2), with the AMS exhibiting higher enhancement

at night compared to day, whereas the FIGAERO-CIMS showed the opposite trend. The reason for the discrepancy may be that continued production of organic aerosol in the airmass below the nighttime forest canopy measured by the AMS is not seen by the FIGAERO-CIMS, due either to lack of mixing between above/below the canopy or that production of organic material by nighttime chemistry was not detectable by iodide-ionization.

Lastly, we observed large variability (on the order of 20% to 30%) in the contribution from each of the subgroups that comprised the total particulate organic mass (figures 6b and 6d). This variability occurred on the timescale of hours driven by the trends in the ambient abundances of each of the subgroups (figure 5). If the diel variability in the ambient abundances of each of the subgroups was driven by their chemical production cycles (as reflected in the gas-phase (figure 4), with which

it is presumably in or close to equilibrium) as opposed to transport (that is, contribution of aerosols with different chemical compositions being advected to the site from elsewhere), the observed diel variability in the contribution from each of the subgroups (figures 6b and 6d) suggests a rapid turnover in the material that comprise the particulate organic mass. That is, the organic material formed at a given time during the day must be lost on comparable timescales, otherwise, the mass fraction it contributes to the total would not change significantly throughout the time of day due to accumulation in the

particle-phase. For instance, we assume two compounds are produced on average at the same rate over the course of a model day, but that their production rates exhibit opposite diel trends, as shown in figure S3a. Their abundances are dictated by the balance between production and loss, as shown in equations (1) and (2):

Eq. (1) $$\frac{dA}{dt} = P_A - \frac{A}{\tau_A}$$

Eq. (2) $$\frac{dB}{dt} = P_B - \frac{B}{\tau_B}$$

, where $P_A$ and $P_B$ are the production rates (s$^{-1}$) of A and B, respectively, and $\tau_A$ and $\tau_B$ are the lifetimes (s) of A and B, respectively. In this case, the lifetimes are with respect to the particle phase component of A and B, and not their overall lifetime in the atmosphere. That is, net repartitioning from the particle phase into the gas-phase due to dilution would be

represented in the lifetimes used in equations (1) and (2).

In this model framework, abundances of A and B eventually reach diurnally-repeating steady states within approximately 5 model days. A and B are analogous to the subgroups defined from the observations above, and the ratio of A (or B) to the sum A+B is analogous to the mass fraction of one of the observational subgroups to the total, as shown in figures 6b and 6d.

We find that if A and B possess lifetimes on the order of a day or longer, then the amplitude and rate of variability in the contribution one to the sum of both, i.e. A/(A+B), are muted compared to the observed relative variations of the subgroups (figures S3b and S3c). Therefore, a rapid loss or short residence time in the particle-phase (on the timescale of hours) is one explanation for the observations highlighted in figures 6b and 6d.

*Lee et al.* [2016] demonstrated that the distinct diel trends exhibited by $C_5$ pON species (presumably derived from isoprene oxidation) and $C_{10}$ pON species (monoterpene-derived) were observed due to their hours-long lifetimes in the particle-phase. Observations during BAECC suggest the lifetime of all organic constituents of the particle-phase, not just organic nitrates,

may be shorter than previously thought. A model to attribute specific chemical reactions and lifetimes to each of the subgroups observed during BAECC requires detailed information on oxidants ($NO_3$, OH, etc.), radicals ($RO_2$, $HO_2$, etc.) and meteorological conditions (boundary layer height, chemical conditions above/below surface layer, etc.) that are beyond the scope of this overview paper.

**3.4 Organic nitrates**

The mass contribution of organic nitrates to the total organic aerosol mass (=pON/(pON+pOC)) exhibited a clear and distinct diel trend, with a maximum of ~0.35 around sunrise and minimum ~0.15 at midday (figure 8a). This was consistent with the trend observed with the same FIGAERO-CIMS coupled package in the Southeast U.S. [*Lee et al.*, 2016]. However, in that study the pON contribution to the total aerosol mass was much lower (~0.05) even though $NO_x$ levels were typically higher

in the summertime in Centreville, AL, particularly at nighttime (figure S4 of [*Lee et al.*, 2016]). Observations at BAECC were also consistent with other observations of unexpectedly high contribution of particle-phase organic nitrates, particularly at nighttime, to the total organic aerosol mass in regions with moderate to low $NO_x$ emissions observed utilizing a network of Aerosol Mass Spectrometers (AMS) by *Kiendler-Scharr et al.* [2016].

It is unclear why the contribution of pON to the total organic aerosol mass was greater above the more pristine (less $NO_x$) boreal forest of Hyytiälä, Finland compared to the more polluted (more $NO_x$) mixed deciduous forest of the Southeast U.S. (figure S1 here and figure S4 of *Lee et al.*, 2016). In addition to $NO_x$, other parameters including monoterpene levels and their speciation (figure S1 of *Ayres et al.* [2015] and *Hakola et al.* [2012]) were different at the two sites. *Kiendlar-Scharr et al.* [2016] attributed the high contribution of ON to total OA to nighttime $NO_3$ radical driven chemistry. However, gas-phase

measurements during BAECC show only 9 ON compounds exhibited a diel trend that would be consistent with $NO_3$-driven chemistry [*Liebmann et al.*, 2018] (table 1), representing at most 10% of the of the total gON mixing ratio (figure 6a). The nighttime pON subgroup constituted at most 50% of the total pON mass in the few hours before sunrise, but its contribution dropped to about 10% at midday (figure 6b). We cannot rule out significant pON production by $NO_3$ chemistry elsewhere, e.g. above the nocturnal surface layer in which the airmass would not be readily accessible to the FIGAERO-CIMS at the top

of tower in the stable nighttime atmosphere. But if faster chemical production of pON in the boreal forest was not the main reason why its contribution to total OA was greater, another possibility is that pON was lost faster relative to pOC in the mixed-deciduous forest with higher temperatures and absolute humidity.

Lastly, we observed ON species possessing 16 or more carbon atoms that belonged to all three subgroups of both the gas and particle phases, as shown in figures 4 and 5, respectively. These compositions are consistent with nitrate dimers of monoterpene oxidation products. The mass contribution of these nitrate dimers to total pON was significant, reaching an average maximum of ~15% at nighttime and a minimum at daytime of ~5% (figure 8a). While there are few, if any reports, of such nitrated dimers in both the gas and particle phases, there is evidence from offline analyses of particle phase SOA of nitrate oligomers [*Nguyen et al.*, 2011]. That we observed some of these dimers enhanced at night is suggestive of a possible role for $NO_3$ radical chemistry. There continues to be debate as to what extent to which $NO_3$ driven oxidation of σ-pinene contributes to SOA formation [*Fry et al.*, 2014; *Kurten et al.*, 2017]. We propose here one way that $NO_3$ radical driven oxidation of σ-pinene could lead to SOA formation, namely through the reaction of the nitrate-derived peroxy radicals undergoing cross reactions with other peroxy radicals, e.g. from ozonolysis or nighttime OH chemistry, to form low volatility nitrated dimers (figure 9). This hypothesis is consistent with the observation of these dimers in the gas-phase, and could be tested in a series of laboratory chamber studies utilizing a FIGAERO-CIMS or similar technique capable of detecting such dimers.

## 4 Conclusions

Hourly measurements using the FIGAERO-CIMS of the abundance and molecular formulae provided a rich view of the organic constituents of the gas and particle phases above a boreal forest during the spring-summer transition season. Reduction of the observations using non-negative matrix factorization revealed that most species in both phases exhibited one of three distinct diel trends, one in which the ambient levels were enhanced at daytime, another during the early morning hours and lastly at nighttime. The mass contribution of each subgroup, comprised of a unique set of compounds and defined by their distinct behavior in time, to the total particulate organic aerosol mass exhibited significant systematic diel variability that is broadly consistent with expectations of daytime photochemistry in the presence of NO and nighttime chemistry dominated by ozonolysis and $NO_3$ chemistry together with diel boundary layer dynamics. Lastly, the contribution of organic nitrates to the total particulate organic mass exhibited a clear nighttime enhancement during BAECC, with a non-negligible contribution from nitrated dimer-like compounds, which may be formed by the reaction between a nitrated organic peroxy radical and non-nitrated peroxy radical. The mass fraction of pON to total OA observed by the same FIGAERO-CIMS was much lower at a temperate forest site in the Southeast U.S. affected by higher nighttime levels of $NO_x$. These observations suggest the rate of loss, more than the production, may have been a key difference between the two sites that determined the contribution of pON to the total organic aerosol burden.

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

Data availability

The FIGAERO-CIMS dataset encompassing all of the organic components of both the gas and particle phases, along with hourly time-dependent desorption profiles of each compound, for the entire 6-week duration of BAECC is several gigabytes. As such, the group at the University of Washington will directly transfer the entire dataset or a version with some of the

dimensions compressed, by means most convenient to interested members of the community. Those who want access should contact the corresponding author, Joel Thornton at thornton@atmos.uw.edu. Datasets of all other observations collected at the SMEAR sites can be accessed here (https://avaa.tdata.fi/web/smart/smear), while being mindful of the terms of use here (https://avaa.tdata.fi/web/smart/smear/terms-of-use).

Acknowledgement

Funding support for the University of Washington was provided by the Department of Energy (DE-SC0006867 and DE-SC0018221). We acknowledge the support from the Academy of Finland Centre of Excellence (project no. 307331), Maj and Tor Nessling funding, the Nordic Centre of Excellence eSTICC (project no. 57001), and the Department of Energy support for AMF2 deployment during BAECC. The help of SMEAR II staff during the field study is gratefully

acknowledged. The authors thank Petri I. R. Keronen for helpful suggestions and contributing the $NO_x$ and $O_3$ observations. Data generated at the SMEAR sites can be accessed here (https://avaa.tdata.fi/web/smart/smear), while being mindful of the terms of use here (https://avaa.tdata.fi/web/smart/smear/terms-of-use).

Competing interests

The authors declare that they have no conflict of interest.

**Table 1: Effective molecular compositions, molecular weights, mixing ratio or mass concentration, and number of species belonging to the four subgroups (daytime, nighttime, morning, and others) of gas-phase organic nitrates ($gON=C_xH_yNO_z$), gas-phase organic carbon ($gOC=C_xH_yN_0O_z$), particulate organic nitrates (pON) and particulate organic carbon (pOC). Corresponding effective statistics for all gas-phase products (gON and gOC) and particle-phase products (pON and pOC) are also shown. The campaign-median mixing ratio of gOC is 62 ppt, which given its effective molecular weight of 121 g $mol^{-1}$ is about 0.3 μg $m^{-3}$.**

|  | Daytime | Nighttime | Morning | Others |
|---|---|---|---|---|
| gas-phase | $C_{4.7}H_{7.8}N_{0.1}O_{4.0}$<br>130 g $mol^{-1}$<br>71.8 ppt<br>n=602 | $C_{9.2}H_{14.9}N_{0.2}O_{3.2}$<br>181 g $mol^{-1}$<br>3.1 ppt<br>n=20 | $C_{7.5}H_{11.7}N_{0.5}O_{4.9}$<br>196 g $mol^{-1}$<br>8.9 ppt<br>n=92 | $C_{8.8}H_{14.4}N_{0.5}O_{4.9}$<br>204 g $mol^{-1}$<br>9.6 ppt<br>n=304 |
| gOC | $C_{4.5}H_{7.5}O_{3.7}$<br>121 g $mol^{-1}$<br>62 ppt<br>n=392 | $C_{9.1}H_{14.7}O_{2.8}$<br>169 g $mol^{-1}$<br>2.5 ppt<br>n=11 | $C_{7.2}H_{11.2}O_{4.2}$<br>182 g $mol^{-1}$<br>4.0 ppt<br>n=33 | $C_{8.5}H_{14.4}O_{4.1}$<br>181 g $mol^{-1}$<br>5.1 ppt<br>n=148 |
| gON | $C_{5.8}H_{9.8}NO_{5.7}$<br>184 g $mol^{-1}$<br>9.4 ppt<br>n=210 | $C_{9.6}H_{15.7}NO_{5.1}$<br>227 g $mol^{-1}$<br>0.6 ppt<br>n=9 | $C_{7.8}H_{12.1}NO_{5.6}$<br>209 g $mol^{-1}$<br>4.9 ppt<br>n=59 | $C_{9.3}H_{14.5}NO_{5.8}$<br>232 g $mol^{-1}$<br>4.5 ppt<br>n=156 |
| particle-phase | $C_{8.5}H_{12.8}N_{0.1}O_{5.8}$<br>212 g $mol^{-1}$<br>0.25 μg $m^{-3}$<br>n=519 | $C_{12.7}H_{20.2}N_{0.9}O_{7.9}$<br>311 g $mol^{-1}$<br>0.031 μg $m^{-3}$<br>n=125 | $C_{8.9}H_{13.5}N_{0.3}O_{5.7}$<br>216 g $mol^{-1}$<br>0.23 μg $m^{-3}$<br>n=332 | $C_{9.4}H_{15.7}N_{0.1}O_{6.1}$<br>231 g $mol^{-1}$<br>$4.0\times10^{-3}$ μg $m^{-3}$<br>n=42 |
| pOC | $C_{8.6}H_{13.1}O_{5.8}$<br>210 g $mol^{-1}$<br>0.23 μg $m^{-3}$<br>n=378 | $C_{16.8}H_{29.2}O_{12.2}$<br>427 g $mol^{-1}$<br>$3.0\times10^{-3}$ μg $m^{-3}$<br>n=28 | $C_{8.5}H_{13.0}O_{5.0}$<br>196 g $mol^{-1}$<br>0.16 μg $m^{-3}$<br>n=151 | $C_{9.6}H_{15.9}O_{6.0}$<br>231 g $mol^{-1}$<br>$3.0\times10^{-3}$ μg $m^{-3}$<br>n=27 |
| pON | $C_{7.4}H_{11.0}NO_{7.1}$<br>227 g $mol^{-1}$<br>0.022 μg $m^{-3}$<br>n=141 | $C_{12.2}H_{19.0}NO_{7.3}$<br>297 g $mol^{-1}$<br>0.028 μg $m^{-3}$<br>n=97 | $C_{9.9}H_{14.8}NO_{7.3}$<br>264 g $mol^{-1}$<br>0.067 μg $m^{-3}$<br>n=181 | $C_{8.2}H_{14.2}NO_{6.3}$<br>228 g $mol^{-1}$<br>$1.0\times10^{-3}$ μg $m^{-3}$<br>n=15 |

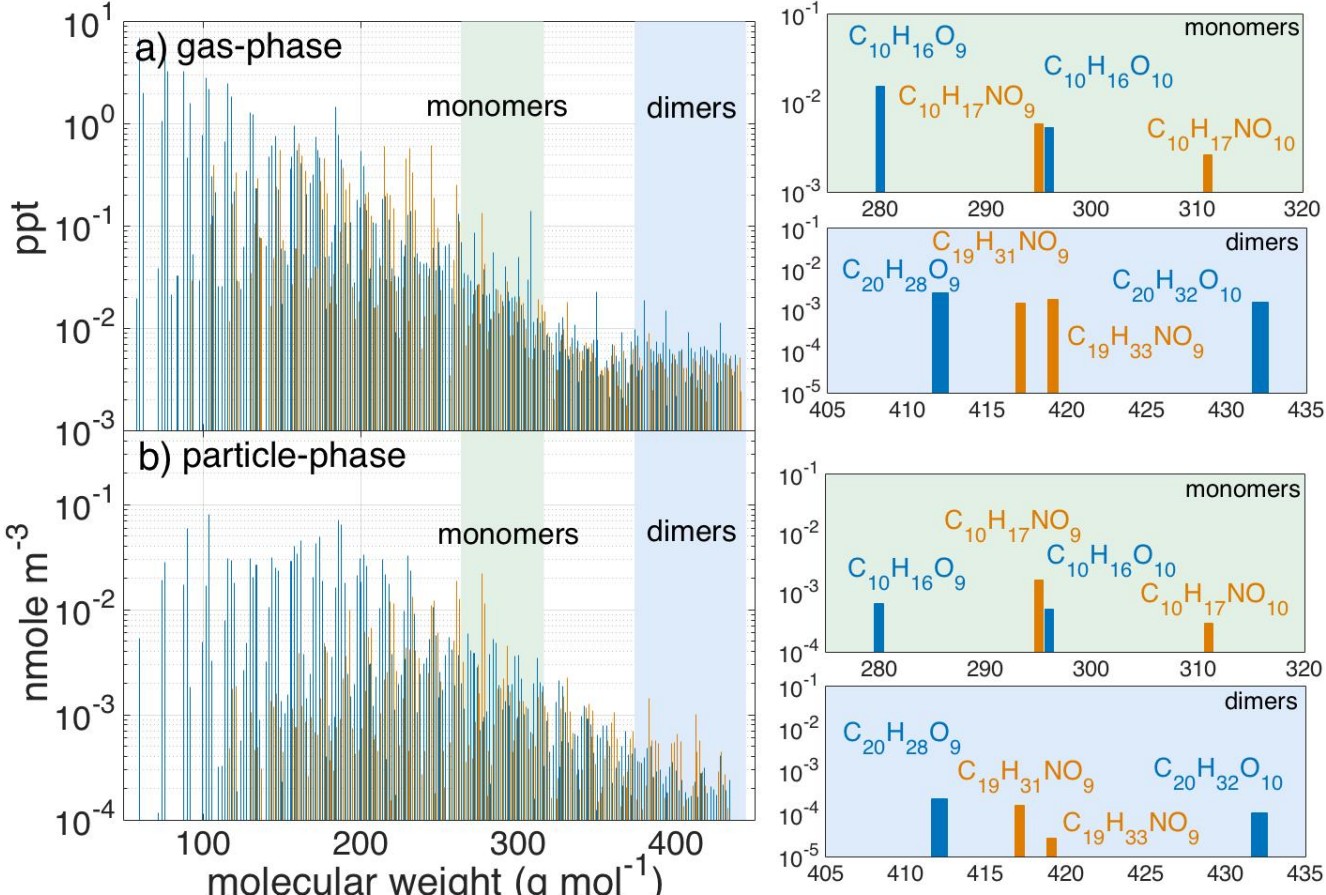

**Figure 1: BAECC-median abundances of the organic nitrate (gold; ON=$C_xH_yNO_z$) and organic carbon (blue; OC=$C_xH_yN_0O_z$) species in the (a) gas and (b) particle phases, plotted as a function of their molecular weights. The shaded green and blue areas in (a) and (b) highlight the molecular weight ranges of monoterpene-derived HOM monomers (nC=10 and nO≥7) and dimers (nC≥16). Sub-panels on the right hand side of (a) and (b) show the abundances of select monomers and dimers of the gas and particle phases, respectively.**

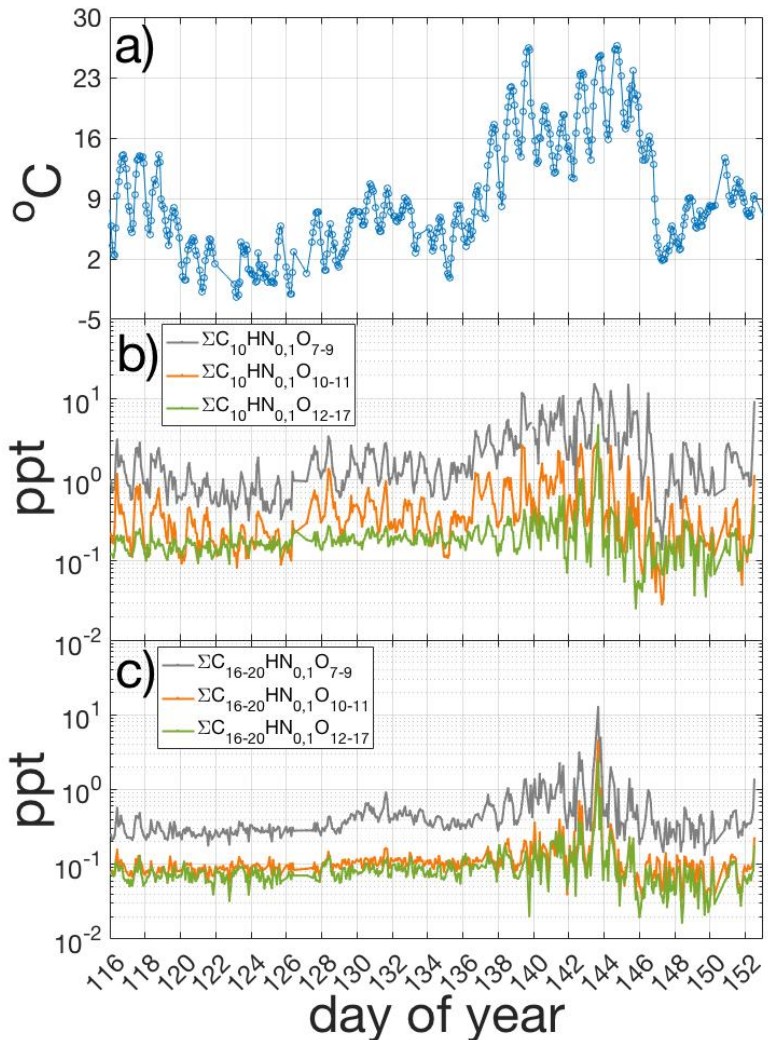

**Figure 2: (a) Ambient temperature, (b) sum of the mixing ratios of three groups of gas-phase HOM monomers with varying ranges of oxygen atom number, and (c) sum of the mixing ratios of three groups of gas-phase HOM dimers with varying ranges of oxygen atom number, all observed during the BAECC campaign in the year 2014. Measurements of (b) and (c) were made with the FIGAERO-CIMS utilizing iodide-adduct ionization.**

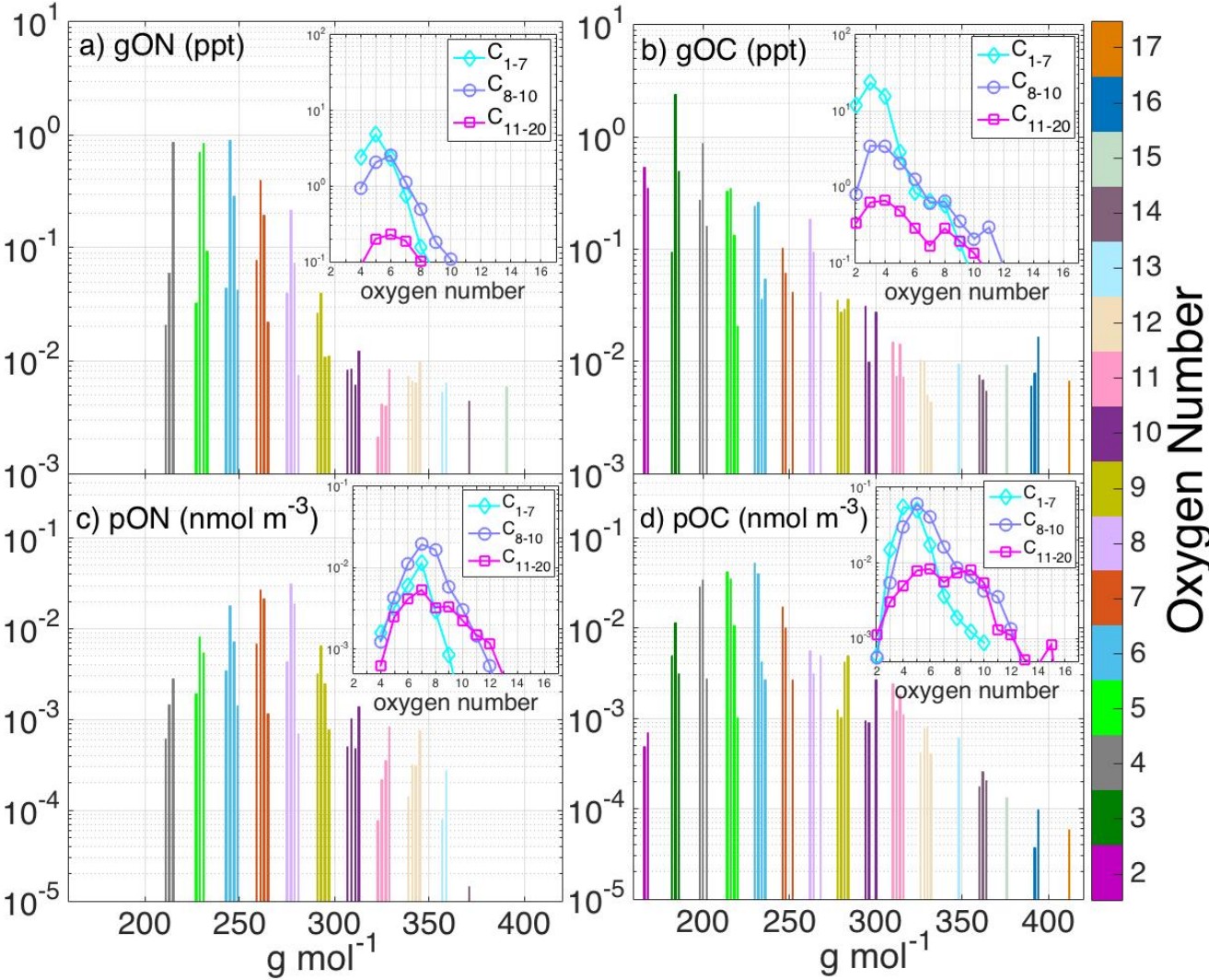

**Figure 3: BAECC-median abundances of (a) gaseous ON, (b) gaseous OC, (c) particulate ON and (d) particulate OC species possessing 10 carbon atoms, as a function of their molecular weights. Colors denote the oxygen atom number of each species. Gas-phase abundance generally decreased with increasing molecular weight, whereas the particle-phase exhibited an approximate bell-shape distribution in abundance. The insets in each panel show the campaign median abundances accounted for by species with 1 to 7, 8 to 10 and 11 to 20 carbon atoms, plotted as a function of their oxygen atom number.**

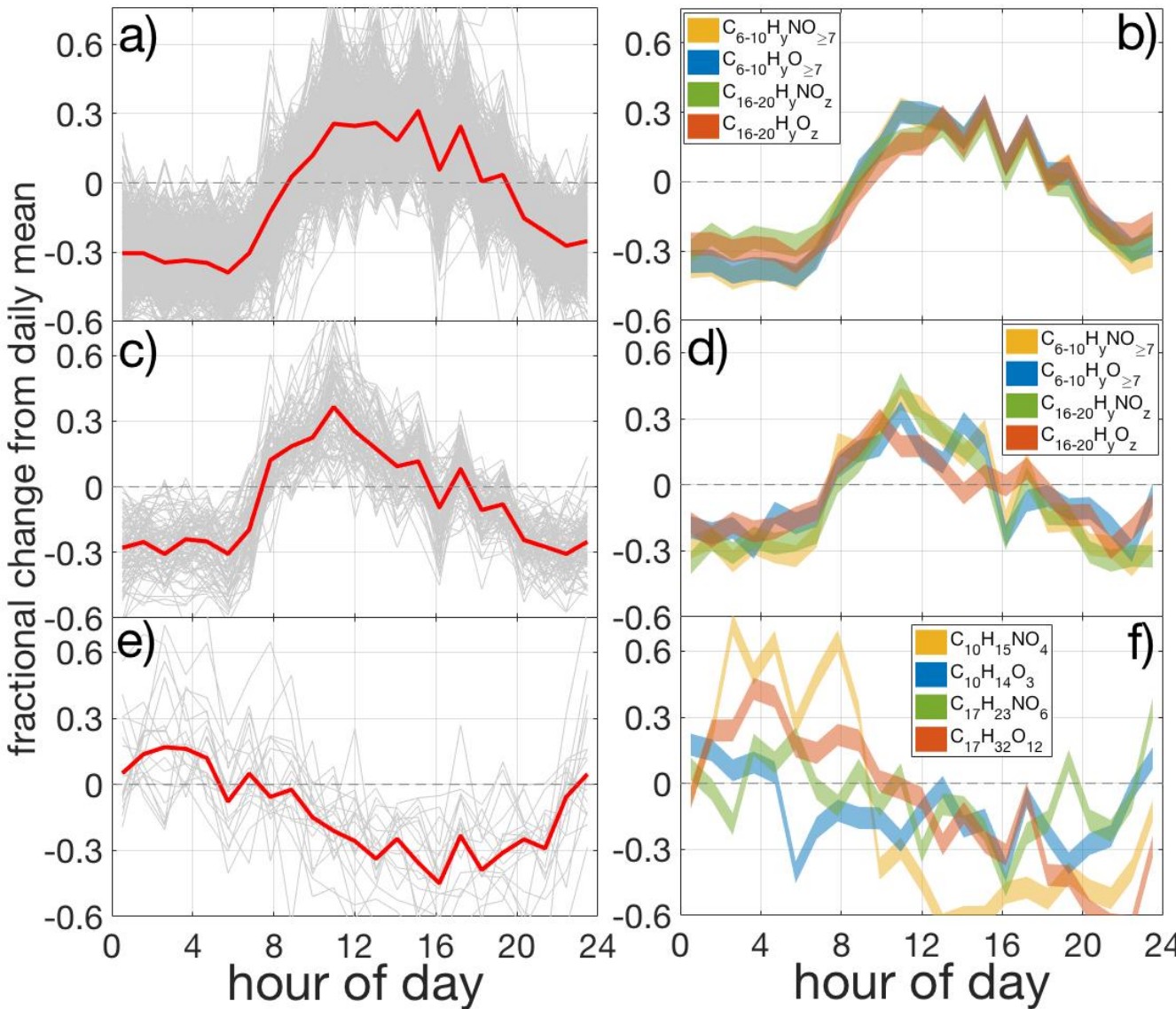

**Figure 4: Fractional change from the daily mean of the organic gas-phase species belonging to the (a) daytime (c) morning, and (e) nighttime subgroups, as categorized using non-negative matrix factorization. Red lines in (a), (c), and (e) represent the means of the species in that subgroup. (b), (d) and (f) show diel trends of all qualifying gON and gOC HOM monomers and dimers corresponding to the three subgroups shown in (a), (c) and (e), respectively. Individual species, as opposed to the mean of a collection of compounds, are shown for the (f) nighttime subgroup, as only 20 species exhibited a nighttime diel trend.**

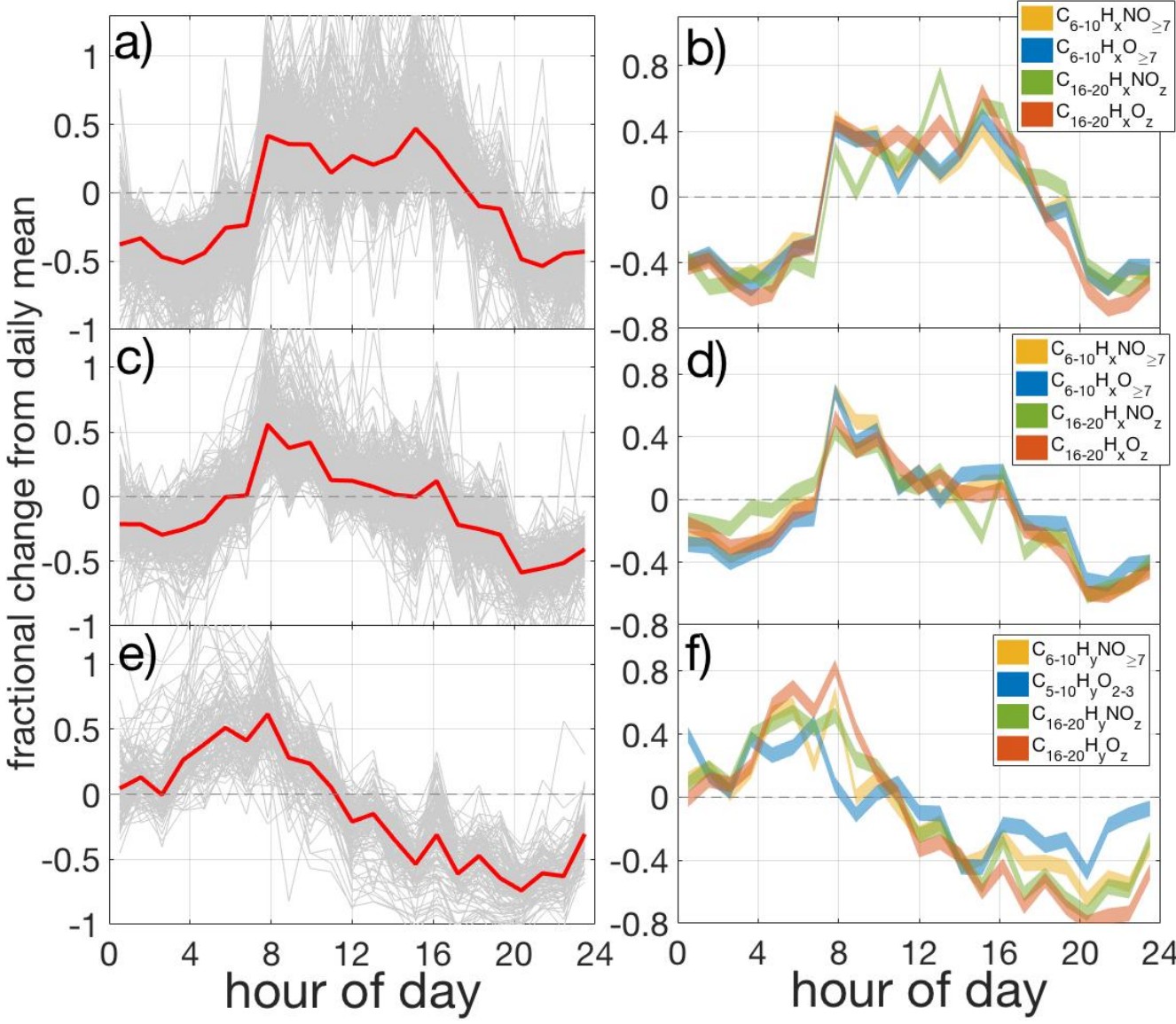

**Figure 5: Same as figure 4, but for the particle-phase. The nighttime subgroup of the particle-phase was dominated by ON species. Only 4 OC compounds with 10 or less carbon atoms exhibited a nighttime diel trend and all possess 3 or less oxygen atoms, as shown (f) in blue.**

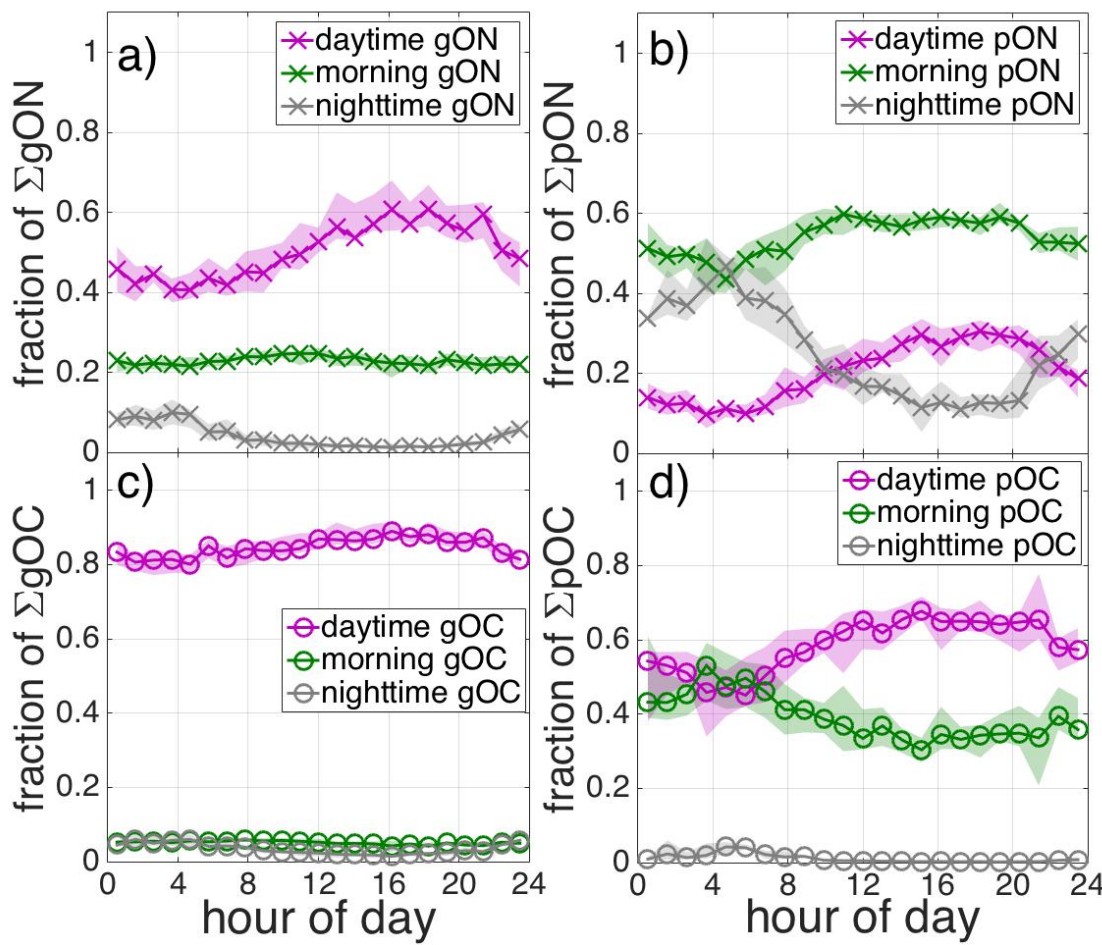

**Figure 6: Mass fractions accounted for by the (purple) daytime, (green) morning and (grey) nighttime subgroups in each of the (a) gON, (b) pON, (c) gOC, and (d) pOC groups, as a function of hour of day. The shaded regions represent 25th and 75th quantile.**

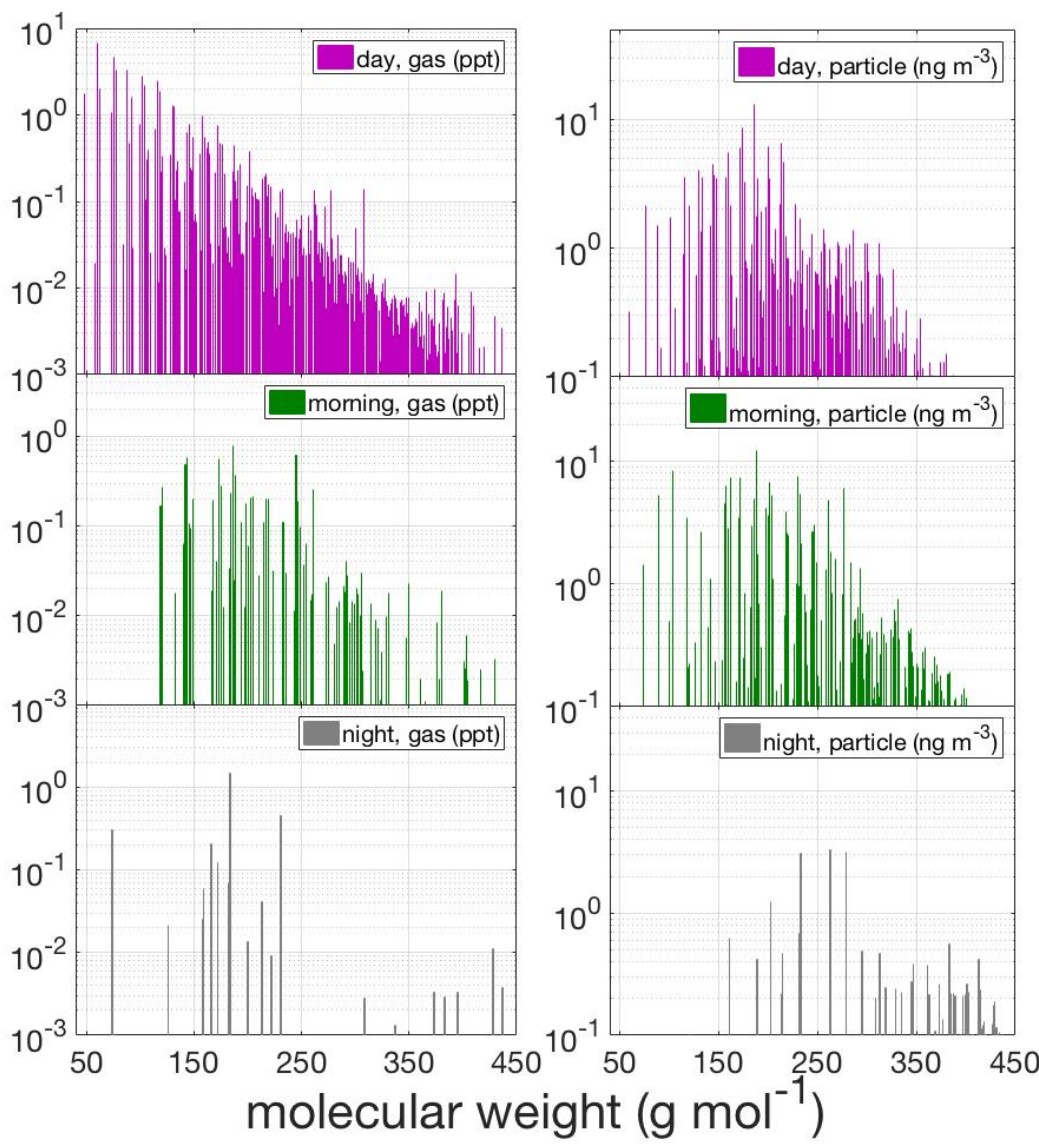

**Figure 7: BAECC-median abundance distributions of the organic species comprising the (left) gas and (right) particle phases of the (top) daytime, (middle) morning, (bottom) nighttime subgroups, plotted as a function of their molecular weights.**

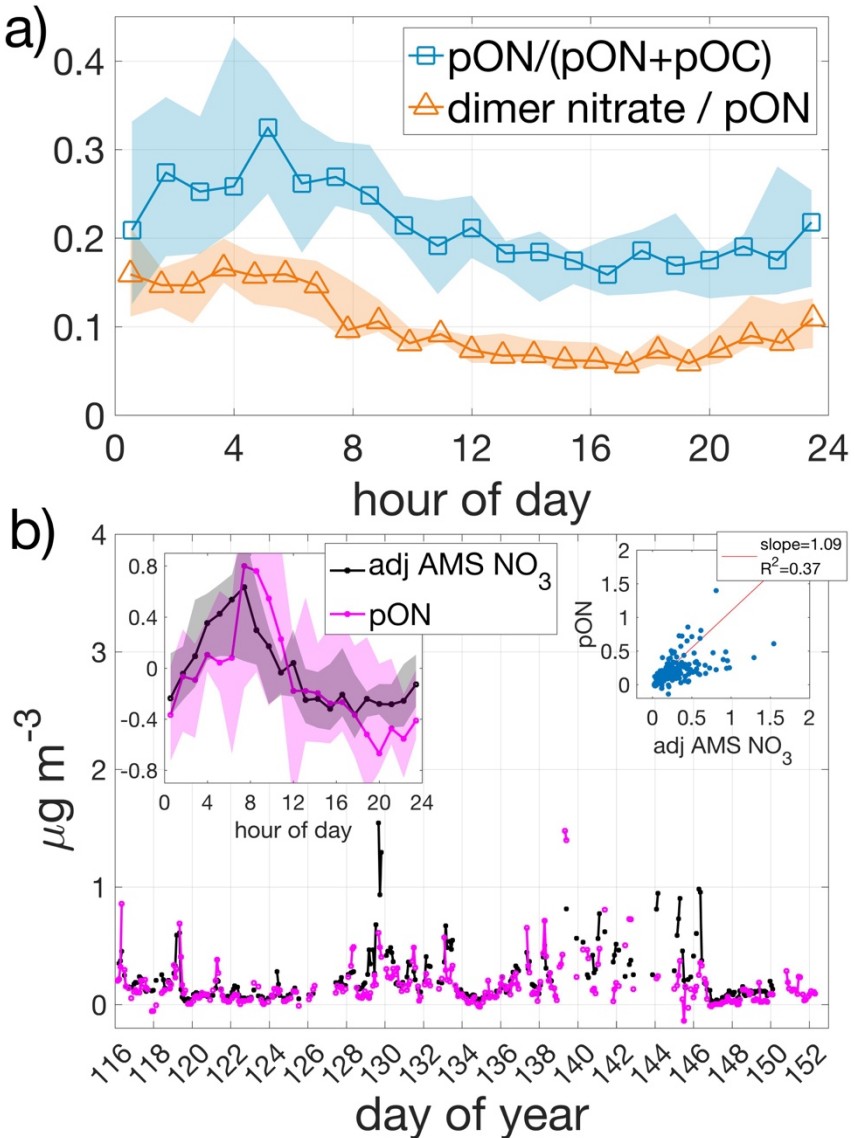

**Figure 8: (a)** Mass fraction of (blue) particulate organic nitrates to the total organic aerosol mass loading, both measured by the FIGAERO-CIMS and mass fraction of (orange) dimer-nitrates to the total organic nitrates, both measured by the FIGAERO-CIMS, as a function of hour of day. **(b)** Mass concentrations of (magenta) particulate organic nitrates measured by FIGAERO-CIMS and (black) total particle $NO_3$ measured by HRToF-AMS but adjusted assuming all of the particle $NO_3$ is composed of organic nitrates with an average molecular weight of 265 g mol$^{-1}$ (i.e. assuming no contribution from inorganic nitrates such that adjusted AMS $NO_3$ = AMS $NO_3 \times 265/62$). Insets in (b) show the comparison between pON and adjusted AMS $NO_3$, and the deviation from their daily respective means as a function of hour of day.

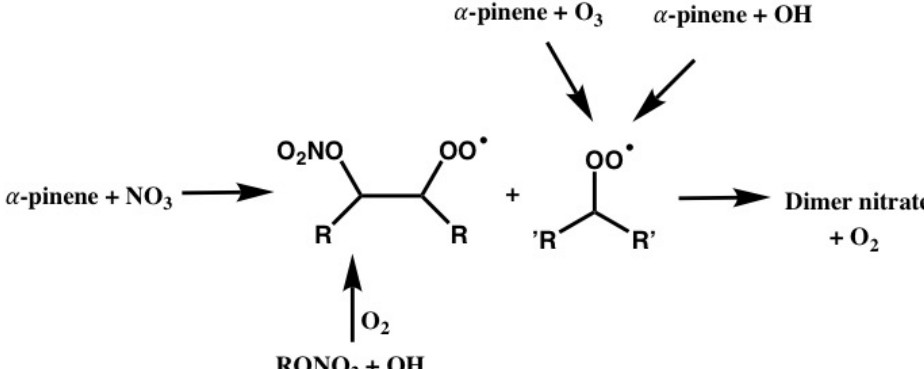

**Figure 9: Schematic of the proposed chemical mechanism responsible for monoterpene dimer nitrates (those possessing 16 or more carbon atoms with a nitrate functional group) observed in the gas and particle phases during BAECC. The nitrated peroxy radical can be produced either by the NO₃-radical oxidation of σ-pinene or by the OH-radical oxidation of σ-pinene nitrate.**

