# Peer review of "Semi-volatile and highly oxygenated gaseous and particulate organic compounds observed above a boreal forest canopy"

_Atmospheric Chemistry and Physics, 2018_

## Referee Comment (RC1) · Anonymous Referee #1 · 5 May 2018

This paper presents a novel and interesting dataset on oxidized organic species contributing to both gas and aerosol phase organic aerosol in a remote boreal forest. The analysis is possible by use of a FIGAERO inlet to monitor gas and aerosol phase separately but using the same I- mass spectrometer, and positive matrix factorization to sift the complex spectra into 3 primary factors with unique diel behaviors. The authors interpret their results as showing a strikingly (considering the remote location and low NOx) large contribution of particulate organic nitrate to the organic aerosol mass concentrations, especially at night. This is consistent with other recent work and thus builds evidence for an increasing role for organonitrates in SOA production. This paper is likely to be of great interest to the SOA research community and I recommend

publication following minor revisions.

General suggestions: 1) since you will ultimately compare the org nitrate contribution to results from Kiendler-Scharr et al around Europe, and SOAS, I suggest to include somewhere in your introduction the average NOx concentration and BVOC composition (is it exclusively a-pinene?) at Hyytiala. Then when you discuss the surprisingly large nitrate contribution, you can point to the differences.

2) in the methods discussion, it sounded like you only ran PMF on the gas phase data. But I think you may have separately done both gas and aerosol? Or did you just use the same groupings as found by the gasphase PMF for both phases to make the later plots? Either way, please clarify in the text.

3) Why does your analysis only include zero or one nitrogen per molecule? Were no molecules with two or more observed, or did you omit them from the analysis?

4) The discussion of variability in figures 6 b and d serving as evidence for the short life-times of some species was confusing to me. I don't see significantly greater variability in those figures compared to e.g. daytime gON in panel a.

5) what is the difference between positive matrix factorization and non-negative? Maybe add a line to the methods explaining the difference and why you chose the latter.

6) can you account for the effect of boundary layer height changes, to help interpret the morning nitrate source?

7) it looks like there is higher pON during the hottest days of your study. Can you comment on this? Can temperature-dependent partitioning be ruled out in explaining any of the diel variation? (Also around p. 8 line 5)

8) P.6 around line 15 you state the yield must be less that 0.5 to explain decreasing Abundance with # of oxygens. Does this assume that whatever does not yield functionalization stays at the same O:C?

9) Does figure 6c mean there is no nighttime o3 chemistry? If all gas phase OC is in the daytime factor? Or is the nighttime factor actually just a nitrate factor and o3 chemistry would be grouped in the daytime OC factor even if there is some at night.

10) figure 7: are the nighttime factors so much sparser MS because there's no autooxidation in there, since nitrates don't need that to be condensable enough?

Minor technical edits: Abstract line 22: mention that this comparator site is in the SE US.

Top of p. 3: suggest to remove the last line of the intro, so you end with the statement of what you add with this work.

P. 4 line 8: " as it is of order a factor of 2"

Line 11: " and the interpretation of these observations"

P. 5 line 7: "approach is that species exhibiting subtle differences...trends may be lumped into"

Line 22: are you talking about levels greater than expected in the particle phase specifically? Clarify

P.6 line 29 " motivates the use of"

P.7 Line 4 and elsewhere: "adhered to" sounds strange to me - how about belonged to?

Line 21: "imply that formation rates.... were sufficiently higher...during the day, consistent with modeling results specific to the SMEAR"

Line 27: "accumulated in the nocturnal "

P. 8 line 10: is the Yan study referenced at the same site & season? Or similar forest type? Suggest to add additional comment specifying, and then in the next lines clarify which study you mean when. "...summer of 2014 reported here observed most

gaseous...whereas those previous measurements near the canopy floor.. summer of 2012 had observed "

P. 9 line 4: I thought nN previously signified average number of and per molecule ? Different meaning here?

P.10 lines 30 and 32: ~0.35 and ~5%: make both fractions or both percents

Line 31" However, in that study the pON"

P.11 1 " BAECC were also consistent with other observations of unexpectedly high..."

Line 6 "was greater above the more pristine"

Around line 25 I'm wondering about the monoterpene distribution & diel cycle at hyytiala

P. 12 line 9 I'm wondering how you assessed the role of boundary layer dynamics

Line 15: ... or difference bvoc mix making sources different, or different temperatures ... might end this is a little more open ended about explanation?

P. 18 table 1: why is only gOC average mixing ratio reported in the caption?

Fig. 1 : why different units on panel b than elsewhere (ng m-3)? Do I interpret the righthand panels correctly to say that all dimer species are more abundant in the gas phase than particle? This seems surprising...

Fig. 2 : are these all gas phase only data?

Fig. 8: explain the "adjustment" a bit more – is this just no3 mass x 265/62?

---

## Referee Comment (RC2) · Anonymous Referee #2 · 6 May 2018

Lee et al. describe aerosol and gas-phase measurements of organic compounds from tall tower located above a boreal forest. The measurements show the diurnal patterns of gas-phase species, measured using an I-CIMS, and particle-phase species, measured using a FIGAERO inlet. The authors find that most gas and particle-phase species exhibit either a morning, daytime, or nighttime enhancement. In the gas-phase, smaller molecules dominated the organic distributions, though highly oxygenated molecules (or HOMs) were observed during the morning and daytime. In the particle phase, HOMs were observed in each diurnal subgroup. Of these compounds, the organic nitrates constituted a significant fraction of the detected organic species, with highest contributions at night. A non-negligible amount of nitrate dimers were observed, which were suspected to be formed by the reaction between NO3RO2 + RO2 radicals.

The results from this study contributes to the evidence that organic nitrate species formed from biogenic VOC oxidation significantly contribute to organic aerosol, especially at night. The results are interesting and well-interpreted, the paper is well written, and the figures are nice and descriptive. I recommend the manuscript for publication provided that the authors address the following very minor comments.

Page 4, lines 26 - It's not clear why NNMF was not applied to raw concentration counts. Is this to give equal weight to all species (i.e., the assumption is that changes in concentrations will be approximately equal across species)? Furthermore, how were the errors estimated? Please clarify.

Page, Lines 8 -12 - I really like this approach for resolving factors, especially as the authors are not trying to over-interpret the data. Can the authors mention how well the variability was explained by the resolved subgroups? Also, what type of residual was left over not explained by NNMF?

Page 5, Lines 21-22 - I'm confused by what the authors are trying to say here. Do the authors mean to say that high abundance masses observed in the gas phase were also observed in the particle phase, but that the presence of these species was unexpected based on volatility? Can the authors give some examples to help orient the reader? This would be useful when interpreting the results in Fig 1.

Page 6, lines 1 - 3. Couldn't the variability also be explained, in part, due to higher emission rates of monoterpenes as a function of temperature?

Page 7, lines 9-11. Do the authors have other data that could show whether the breakup of the nocturnal boundary layer contributed to the trends observed here? Were there vertically resolved measurements (e.g. temperature, RH, etc) that support the presented of a nocturnal layer below the tower? I realize that this will not change

the interpretation of gas and particle phase correlations, but it would be interesting to know if the morning diel pattern is dominated by sudden burst of species produced during the night time, or by a sudden burst in oxygenated species once photochemistry kicked in.

Page 9, Lines 12-23. Is it reasonable to infer that the agreement between the AMS (located below the forest canopy) and FIGAERO CIMS (located above the forest canopy) in pON provides evidence that that the tall tower was within the nocturnal boundary layer?

Figure 3: This figure is great and conveys a lot of information. Can the authors comment on what appears to be a bi-modal distribution in the C11-C20 compounds? There appears to be two peaks in the nO distributions, with one peaking around 5-6 oxygens, and the other peaking at 8-10 oxygens. Is this related to carbon number, or is this explained more readily by other processes (auto-oxidation of dimers)?

---

## Author Comment (AC1) · 13 Jul 2018

Anonymous Referee #1 This paper presents a novel and interesting dataset on oxidized organic species contributing to both gas and aerosol phase organic aerosol in a remote boreal forest. The analysis is possible by use of a FIGAERO inlet to monitor gas and aerosol phase separately but using the same I- mass spectrometer, and positive matrix factorization to sift the complex spectra into 3 primary factors with unique diel behaviors. The authors interpret their results as showing a strikingly (considering the remote location and low NOx) large contribution of particulate organic nitrate to the organic aerosol mass concentrations, especially at night. This is consistent with other

recent work and thus builds evidence for an increasing role for organonitrates in SOA production. This paper is likely to be of great interest to the SOA research community and I recommend publication following minor revisions.

We thank the reviewer for their detailed comments and questions. They clearly reflect the time and attention paid to the review.

General suggestions: 1) since you will ultimately compare the org nitrate contribution to results from Kiendler-Scharr et al around Europe, and SOAS, I suggest to include somewhere in your introduction the average NOx concentration and BVOC composition (is it exclusively a-pinene?) at Hyytiala. Then when you discuss the surprisingly large nitrate contribution, you can point to the differences.

Since the mixing ratios of monoterpenes vary greatly as a function of time of day just like NOx (which is shown in figure S1), we now cite two papers that include figures showing monoterpenes as a function of time in the section that discusses organic nitrates so interested readers can find the relevant figures and information (lines 8-9, page 11). Based on these references, we estimate that in terms of OH and ozone reactivity, the two dominant monoterpenes are ïĄ̧ą-pinene and -3-carene, where the latter is important but on a more episodic basis.

2) in the methods discussion, it sounded like you only ran PMF on the gas phase data. But I think you may have separately done both gas and aerosol? Or did you just use the same groupings as found by the gasphase PMF for both phases to make the later plots? Either way, please clarify in the text.

Yes, NNMF was performed separately for each phase. We clarified the statement on lines 29-30, page 4. We have not run the factorization on the combined gas+particle phase timeseries. This could be of interest and will likely better fit into a forthcoming manuscript on factorization using the time-resolved thermogram information.

3) Why does your analysis only include zero or one nitrogen per molecule? Were no

molecules with two or more observed, or did you omit them from the analysis?

There were 27 organic di-nitrates (nN=2) identified in the mass spectra recorded during the BAECC campaign. Though the presence of these di-nitrates are interesting in and of themselves, they represented a small mass fraction of the total organic aerosol mass as measured by the FIGAERO-CIMS. Additionally, there is often a fair amount of uncertainty in attribution of signal to di-nitrates because the mass spectral signals often overlap with more abundant non-nitrogen containing compounds. As such, they were omitted from this bulk analysis.

4) The discussion of variability in figures 6 b and d serving as evidence for the short lifetimes of some species was confusing to me. I don't see significantly greater variability in those figures compared to e.g. daytime gON in panel a.

Abundances of gaseous organic material could exhibit large diel variability given that they are byproducts of oxidants and BVOC, both of which exhibit large diel variabilities. Abundance of organic material in the particle-phase is governed by the integral of production and loss, where potentially the largest loss term is from physical removal of the particle, for example, due to wet deposition or horizontal transport, both on the order of days and lacking a diurnal pattern. We observed, as the reviewer also notes, a distinct diel variability in the contribution of each subgroup to the total OA (figure 6). This could only occur if the particulate organic material also has a short (hours-long) lifetime or if there is a large production rate that is highly localized. Otherwise, their diel variability becomes dampened, or less distinct. We have now clarified the statements in the first paragraph of page 10 to reflect this discussion.

5) what is the difference between positive matrix factorization and non-negative? Maybe add a line to the methods explaining the difference and why you chose the latter.

We have now revised paragraph 3 of page 4. Briefly, NNMF and PMF are very similar, but NNMF allows for "0" mass, and is part of the Matlab software package.

Interactive
comment

6) can you account for the effect of boundary layer height changes, to help interpret the morning nitrate source?

The boundary layer height was measured by a ceilometer during BAECC. However, without FIGAERO-CIMS measurements above and below the boundary layer, the boundary layer height measurement alone does little to shed light on its effect on the observed diel trends. We also note that the boundary layer height (or cloud base height, which is what is measured by the ceilometer) does not routinely fall below the measurement height at the top of the tower. The same argument applies to the lack of mixing between above and below the forest canopy at nighttime. Without knowing the distribution of organic compounds detected by the FIGAERO-CIMS below the canopy, it is difficult to comment on the extent of influence that vertical mixing had on the observed diel trends. We state now more clearly on lines 7-9 (page 10) that boundary layer dynamics can have an effect on the observed diel trends.

7) it looks like there is higher pON during the hottest days of your study. Can you comment on this? Can temperature-dependent partitioning be ruled out in explaining any of the diel variation? (Also around p. 8 line 5)

Figure 2 shows the mixing ratio of gaseous HOM monomers and dimers. We now clarify that in the figure caption. And yes, ambient temperature affects emission rates of monoterpenes (and likely soil NOx emissions), and often associated with stagnation (high pressure), therefore, it can be expected that absolute concentrations of organic nitrates would increase with ambient temperature.

8) P.6 around line 15 you state the yield must be less that 0.5 to explain decreasing Abundance with # of oxygens. Does this assume that whatever does not yield functionalization stays at the same O:C?

Yes. We have now clarified that statement in the last paragraph of page 6.

9) Does figure 6c mean there is no nighttime o3 chemistry? If all gas phase OC is

in the daytime factor? Or is the nighttime factor actually just a nitrate factor and o3 chemistry would be grouped in the daytime OC factor even if there is some at night.

The fact that much of the mass of non-nitrate gaseous organic material (shown on figure 6c) exhibits a daytime enhanced trend likely means that much of it is produced during daytime when BVOC emission rates are at their highest and that these oxidation products remain throughout the night when production has slowed. But, the above statement does not mean nighttime ozone chemistry is absent, only that in a relative sense there is an enhancement in OC above the canopy due to daytime emissions and chemistry. Given that factorization will pick out groups with large relative variance, it is possible that the "nighttime factor" is dominated by a relatively larger nitrate variance. Ozone chemistry occurs day and night, and thus has a less pronounced local or diel variability, likely causing its contribution to be split into multiple factors. These issues are problematic for interpretation of results from any factorization approach, as it is a statistical pattern not necessarily a causal pattern.

10) figure 7: are the nighttime factors so much sparser MS because there's no autoox- idation in there, since nitrates don't need that to be condensable enough?

Slower rate of auto-oxidation due to lower ambient temperatures could be a factor why the FIGAERO-CIMS observed fewer organic species that belonged to the nighttime subgroup. But, see above for other possible effects of factorization artificially masking nighttime ozone chemistry. If the temperature is high enough, nighttime ozonolysis should lead to autoxidation. Determining the relative fates of RO2 (auto-oxidation, reaction with NO, RO2, or HO2) may be informative, and should be pursued in the future. However, nitrates are not necessarily more condensable, in fact, a nitrate with the same O/C as a non-nitrate (e.g. $C_{10}H_{15}O_8N$ $C_{10}H_{16}O_8$), is expected (from group contribution estimates) to have a higher saturation vapor pressure, and thus be "less condensable".

Minor technical edits: Abstract line 22: mention that this comparator site is in the SE

US.

Done

Top of p. 3: suggest to remove the last line of the intro, so you end with the statement of what you add with this work.

Done

P.4line8: "asitisoforderafactorof2"

Done

Line 11: " and the interpretation of these observations"

Done

P. 5 line 7: "approach is that species exhibiting subtle differences...trends may be lumped into"

Done

Line 22: are you talking about levels greater than expected in the particle phase specifically? Clarify

We have clarified the statement on page 5, line 27-28.

P.6 line 29 " motivates the use of"

Done

P.7 Line 4 and elsewhere: "adhered to" sounds strange to me - how about belonged to?

Done

Line 21: "imply that formation rates.... were sufficiently higher...during the day, consistent with modeling results specific to the SMEAR" Done

Line 27: "accumulated in the nocturnal " Done

P. 8 line 10: is the Yan study referenced at the same site & season? Or similar forest type? Suggest to add additional comment specifying, and then in the next lines clarify which study you mean when. "...summer of 2014 reported here observed most gaseous...whereas those previous measurements near the canopy floor.. summer of 2012 had observed "

Done

P. 9 line 4: I thought nN previously signified average number of and per molecule ? Different meaning here?

The effective atom numbers are mass-weighted. If a subgroup is composed of only non-nitrates, the effective nN is 0. If all nitrates, then nN=1. If 50/50 by mass, then nN=0.5

P.10 lines 30 and 32: âĹij0.35 and âĹij5%: make both fractions or both percents

Done

Line 31" However, in that study the pON"

Done

P.11 1 " BAECC were also consistent with other observations of unexpectedly high..."

Done

Line 6 "was greater above the more pristine"

Done

Around line 25 I'm wondering about the monoterpene distribution & diel cycle at hyytiala

To our knowledge, there were no speciated measurements of monoterpenes during

BAECC. The study by Hakola et al., [2012 ACP] cited in the manuscript is the most recent work at the site.

P. 12 line 9 I'm wondering how you assessed the role of boundary layer dynamics

See above discussion in response to an earlier comment. Normalizing to the total OA focuses changes in relative composition not absolute abundance to avoid a direct effect of boundary layer height changes.

Line 15: ... or difference bvoc mix making sources different, or different temperatures ... might end this is a little more open ended about explanation?

With the added sentence near the beginning of this paragraph, we believe we have now conveyed that there are many differences between the two sites that need to be investigated further.

P. 18 table 1: why is only gOC average mixing ratio reported in the caption?

That was an example to show how ppt converts to ug m-3.

Fig. 1 : why different units on panel b than elsewhere (ng m-3)? Do I interpret the righthand panels correctly to say that all dimer species are more abundant in the gas phase than particle? This seems surprising...

Figure 1a shows gas-phase in units of ppt. Figure 1b shows particle-phase in units of nmole m-3.

Fig. 2 : are these all gas phase only data?

We have revised the figure 2 caption to clarify that the data in (b) and (c) are both of the gas-phase.

Fig. 8: explain the "adjustment" a bit more – is this just no3 mass x 265/62?

Yes. Figure 8 caption has been revised to show that more clearly.

Anonymous Referee #2

Lee et al. describe aerosol and gas-phase measurements of organic compounds from tall tower located above a boreal forest. The measurements show the diurnal patterns of gas-phase species, measured using an I-CIMS, and particle-phase species, measured using a FIGAERO inlet. The authors find that most gas and particle-phase species exhibit either a morning, daytime, or nighttime enhancement. In the gas-phase, smaller molecules dominated the organic distributions, though highly oxygenated molecules (or HOMs) were observed during the morning and daytime. In the particle phase, HOMs were observed in each diurnal subgroup. Of these compounds, the organic nitrates constituted a significant fraction of the detected organic species, with highest contributions at night. A non-negligible amount of nitrate dimers were observed, which were suspected to be formed by the reaction between NO3RO2 + RO2 radicals. The results from this study contributes to the evidence that organic nitrate species formed from biogenic VOC oxidation significantly contribute to organic aerosol, especially at night. The results are interesting and well-interpreted, the paper is well written, and the figures are nice and descriptive. I recommend the manuscript for publication provided that the authors address the following very minor comments.

The authors greatly appreciate the reviewer for their detailed comments and suggestions.

Page 4, lines 26 - It's not clear why NNMF was not applied to raw concentration counts. Is this to give equal weight to all species (i.e., the assumption is that changes in concentrations will be approximately equal across species)? Furthermore, how were the errors estimated? Please clarify.

We perform NNMF on the deviation from the daily mean of each species, so regardless of whether NNMF is performed using mixing ratio, mass concentration, or raw signal counts, each species is effectively treated with equal weight in this approach. Factorization will create groups that explain the largest fraction of total variance. The dynamic range of CIMS means that a few very large peaks will often dominate and mask other possible components. We have now clarified that on line 28, page 4. We

do not account for uncertainties, though the precision error is negligible, and we restrict the approach to produce only a few of the dominant factors. There is likely a calibration uncertainty that is large, but difficult to quantify for individual species, hence, another motivation for giving all components an equal weight.

Page, Lines 8 -12 - I really like this approach for resolving factors, especially as the authors are not trying to over-interpret the data. Can the authors mention how well the variability was explained by the resolved subgroups? Also, what type of residual was left over not explained by NNMF?

We state that each species that was deemed as belonging to a given subgroup exhibited a correlation coefficient ($R2$) better than 0.45 with that subgroup's diel trend determined by NNMF (lines 7-9, page 5). The species that did not exhibit a sufficiently distinct enough diel trend, or the "others" subgroup, are effectively the residual.

Page 5, Lines 21-22 - I'm confused by what the authors are trying to say here. Do the authors mean to say that high abundance masses observed in the gas phase were also observed in the particle phase, but that the presence of these species was unexpected based on volatility? Can the authors give some examples to help orient the reader? This would be useful when interpreting the results in Fig 1. We have now clarified that statement on page 5, lines 21-24. Organic compounds typically exhibit an approximate bell-shape distribution in the particle-phase, with the most abundant organic materials possessing molecular weight of ∼220 g/mol. The exception appears to be the 50 or so species at the low molecular weight (∼125) end that are nearly as abundant as the material with higher molecular weight. We assume these compounds likely resulted from thermal fragmentation of higher molecular weight material (as described in Lopez-Hilfiker et al ACP 2015).

Page 6, lines 1 - 3. Couldn't the variability also be explained, in part, due to higher emission rates of monoterpenes as a function of temperature? Yes, we have now clarified that comment on page 6, line 8.

Page 7, lines 9-11. Do the authors have other data that could show whether the breakup of the nocturnal boundary layer contributed to the trends observed here? Were there vertically resolved measurements (e.g. temperature, RH, etc) that support the presented of a nocturnal layer below the tower? I realize that this will not change the interpretation of gas and particle phase correlations, but it would be interesting to know if the morning diel pattern is dominated by sudden burst of species produced during the night time, or by a sudden burst in oxygenated species once photochemistry kicked in.

There are vertical profile temperature measurements from another tower at the same site that, along with published reports (i.e. Zha et al., 2018; Schobesberger et al., 2016), show that there is a de-coupling of air above and below the forest canopy at nighttime when the vertical mixing becomes relatively stagnant. However, without FIGAERO-CIMS measurements above and below the canopy, the influence of mixing on the observed diel trends is difficult to definitively conclude.

Page 9, Lines 12-23. Is it reasonable to infer that the agreement between the AMS (located below the forest canopy) and FIGAERO CIMS (located above the forest canopy) in pON provides evidence that that the tall tower was within the nocturnal boundary layer?

That is a keen observation by the reviewer. We strongly suspect there is strong nighttime decoupling of the air near the surface from above the canopy. That the diel trends of AMS NO3 below the canopy and FIGAERO-CIMS pON above the canopy appear similar is likely due to the fact that organic nitrate production (due to NO3-radical-driven chemistry) and partitioning to the particle-phase (colder ambient temperature) are both relatively stronger at night compared to day.

Figure 3: This figure is great and conveys a lot of information. Can the authors comment on what appears to be a bi-modal distribution in the C11-C20 compounds? There appears to be two peaks in the nO distributions, with one peaking around 5-6 oxygens,

and the other peaking at 8-10 oxygens. Is this related to carbon number, or is this explained more readily by other processes (auto-oxidation of dimers)?

That is a great pickup on the part of the reviewer. We have added a statement on this on page 6, lines 27-33 noting this observation. There does appear to be a noticeable drop in abundance of C11 to C20 compounds (insets of figure 1b and 1d) that possess 7 oxygen atoms compared to those with 5-6 and 8-10. We speculate that such a behavior is due to the combined effects of OH oxidation or ozonolysis and auto-oxidation leading to sequential addition of O2 that possibly do not favor the formation of O7 species, as well as volatilities of the resulting products that generally tends to decrease with increasing oxygen atom number. A detailed chemical model with observations from controlled laboratory experiments is needed to make a more informed assessment.

---

## Author Response (AR1)

This paper presents a novel and interesting dataset on oxidized organic species contributing to both gas and aerosol phase organic aerosol in a remote boreal forest. The analysis is possible by use of a FIGAERO inlet to monitor gas and aerosol phase
5  separately but using the same I- mass spectrometer, and positive matrix factorization to sift the complex spectra into 3 primary factors with unique diel behaviors. The authors interpret their results as showing a strikingly (considering the remote location and low NOx) large contribution of particulate organic nitrate to the organic aerosol mass concentrations, especially at night. This is consistent with other recent work and
10  thus builds evidence for an increasing role for organonitrates in SOA production. This paper is likely to be of great interest to the SOA research community and I recommend publication following minor revisions.

*We thank the reviewer for their detailed comments and questions. They clearly reflect the time and attention paid to the review.*

General suggestions: 1) since you will ultimately compare the org nitrate contribution to results from Kiendler-Scharr et al around Europe, and SOAS, I suggest to include somewhere in your introduction the average NOx concentration and BVOC composition (is it exclusively a-pinene?) at Hyytiala. Then when you discuss the
20  surprisingly large nitrate contribution, you can point to the differences.

*Since the mixing ratios of monoterpenes vary greatly as a function of time of day just like NOx (which is shown in figure S1), we now cite two papers that include figures showing monoterpenes as a function of time in the section that discusses organic nitrates so interested readers can find the relevant figures and information (lines 8-9,*
25  *page 11). Based on these references, we estimate that in terms of OH and ozone reactivity, the two dominant monoterpenes are α-pinene and Δ-3-carene, where the latter is important but on a more episodic basis.*

2) in the methods discussion, it sounded like you only ran PMF on the gas phase data.

But I think you may have separately done both gas and aerosol? Or did you just use the same groupings as found by the gasphase PMF for both phases to make the later plots? Either way, please clarify in the text.

*Yes, NNMF was performed separately for each phase. We clarified the statement on lines 29-30, page 4. We have not run the factorization on the combined gas+particle phase timeseries. This could be of interest and will likely better fit into a forthcoming manuscript on factorization using the time-resolved thermogram information.*

3) Why does your analysis only include zero or one nitrogen per molecule? Were no molecules with two or more observed, or did you omit them from the analysis?

*There were 27 organic di-nitrates (nN=2) identified in the mass spectra recorded during the BAECC campaign. Though the presence of these di-nitrates are interesting in and of themselves, they represented a small mass fraction of the total organic aerosol mass as measured by the FIGAERO-CIMS. Additionally, there is often a fair amount of uncertainty in attribution of signal to di-nitrates because the mass spectral signals often overlap with more abundant non-nitrogen containing compounds. As such, they were omitted from this bulk analysis.*

4) The discussion of variability in figures 6 b and d serving as evidence for the short lifetimes of some species was confusing to me. I don't see significantly greater variability in those figures compared to e.g. daytime gON in panel a.

*Abundances of gaseous organic material could exhibit large diel variability given that they are byproducts of oxidants and BVOC, both of which exhibit large diel variabilities. Abundance of organic material in the particle-phase is governed by the integral of production and loss, where potentially the largest loss term is from physical removal of the particle, for example, due to wet deposition or horizontal transport, both on the order of days and lacking a diurnal pattern. We observed, as the reviewer also notes, a distinct diel variability in the contribution of each subgroup to the total OA (figure 6). This could only occur if the particulate organic material*

*also has a short (hours-long) lifetime or if there is a large production rate that is highly localized. Otherwise, their diel variability becomes dampened, or less distinct. We have now clarified the statements in the first paragraph of page 10 to reflect this discussion.*

5) what is the difference between positive matrix factorization and non-negative? Maybe add a line to the methods explaining the difference and why you chose the latter.

*We have now revised paragraph 3 of page 4. Briefly, NNMF and PMF are very similar, but NNMF allows for "0" mass, and is part of the Matlab software package.*

6) can you account for the effect of boundary layer height changes, to help interpret the morning nitrate source?

*The boundary layer height was measured by a ceilometer during BAECC. However, without FIGAERO-CIMS measurements above and below the boundary layer, the boundary layer height measurement alone does little to shed light on its effect on the observed diel trends. We also note that the boundary layer height (or cloud base height, which is what is measured by the ceilometer) does not routinely fall below the measurement height at the top of the tower. The same argument applies to the lack of mixing between above and below the forest canopy at nighttime. Without knowing the distribution of organic compounds detected by the FIGAERO-CIMS below the canopy, it is difficult to comment on the extent of influence that vertical mixing had on the observed diel trends. We state now more clearly on lines 7-9 (page 10) that boundary layer dynamics can have an effect on the observed diel trends.*

7) it looks like there is higher pON during the hottest days of your study. Can you comment on this? Can temperature-dependent partitioning be ruled out in explaining any of the diel variation? (Also around p. 8 line 5)

*Figure 2 shows the mixing ratio of gaseous HOM monomers and dimers. We now clarify that in the figure caption. And yes, ambient temperature affects emission rates of monoterpenes (and likely soil NOx emissions), and often associated with stagnation (high pressure), therefore, it can be expected that absolute concentrations of organic*
5   *nitrates would increase with ambient temperature.*

8) P.6 around line 15 you state the yield must be less that 0.5 to explain decreasing Abundance with # of oxygens. Does this assume that whatever does not yield functionalization stays at the same O:C?

10  *Yes. We have now clarified that statement in the last paragraph of page 6.*

9) Does figure 6c mean there is no nighttime o3 chemistry? If all gas phase OC is in the daytime factor? Or is the nighttime factor actually just a nitrate factor and o3 chemistry would be grouped in the daytime OC factor even if there is some at night.

15  *The fact that much of the mass of non-nitrate gaseous organic material (shown on figure 6c) exhibits a daytime enhanced trend likely means that much of it is produced during daytime when BVOC emission rates are at their highest and that these oxidation products remain throughout the night when production has slowed. But, the above statement does not mean nighttime ozone chemistry is absent, only that in a*
20  *relative sense there is an enhancement in OC above the canopy due to daytime emissions and chemistry. Given that factorization will pick out groups with large relative variance, it is possible that the "nighttime factor" is dominated by a relatively larger nitrate variance. Ozone chemistry occurs day and night, and thus has a less pronounced local or diel variability, likely causing its contribution to be*
25  *split into multiple factors. These issues are problematic for interpretation of results from any factorization approach, as it is a statistical pattern not necessarily a causal pattern.*

10) figure 7: are the nighttime factors so much sparser MS because there's no autooxidation in there, since nitrates don't need that to be condensable enough?

*Slower rate of auto-oxidation due to lower ambient temperatures could be a factor why the FIGAERO-CIMS observed fewer organic species that belonged to the nighttime subgroup. But, see above for other possible effects of factorization artificially masking nighttime ozone chemistry. If the temperature is high enough, nighttime ozonolysis should lead to autoxidation. Determining the relative fates of RO2 (auto-oxidation, reaction with NO, RO2, or HO2) may be informative, and should be pursued in the future. However, nitrates are not necessarily more condensable, in fact, a nitrate with the same O/C as a non-nitrate (e.g. $C_{10}H_{15}O_8N$ $C_{10}H_{16}O_8$), is expected (from group contribution estimates) to have a higher saturation vapor pressure, and thus be "less condensable".*

Minor technical edits: Abstract line 22: mention that this comparator site is in the SE US.

*Done*

Top of p. 3: suggest to remove the last line of the intro, so you end with the statement of what you add with this work.

*Done*

P.4line8: "asitisoforderafactorof2"

*Done*

Line 11: " and the interpretation of these observations"

*Done*

P. 5 line 7: "approach is that species exhibiting subtle differences...trends may be lumped into"

5  *Done*

Line 22: are you talking about levels greater than expected in the particle phase specifically? Clarify

*We have clarified the statement on page 5, line 27-28.*

P.6 line 29 " motivates the use of"

*Done*

P.7 Line 4 and elsewhere: "adhered to" sounds strange to me - how about belonged
15  to?

*Done*

Line 21: "imply that formation rates.... were sufficiently higher...during the day, consis- tent with modeling results specific to the SMEAR"

20  *Done*

Line 27: "accumulated in the nocturnal "

*Done*

P. 8 line 10: is the Yan study referenced at the same site & season? Or similar forest type? Suggest to add additional comment specifying, and then in the next lines clarify which study you mean when. "...summer of 2014 reported here observed most gaseous...whereas those previous measurements near the canopy floor.. summer of 2012 had observed "

*Done*

P. 9 line 4: I thought nN previously signified average number of and per molecule ? Different meaning here?

*The effective atom numbers are mass-weighted. If a subgroup is composed of only non-nitrates, the effective nN is 0. If all nitrates, then nN=1. If 50/50 by mass, then nN=0.5*

P.10 lines 30 and 32: ~0.35 and ~5%: make both fractions or both percents

*Done*

Line 31" However, in that study the pON"

*Done*

P.11 1 " BAECC were also consistent with other observations of unexpectedly high..."

*Done*

Line 6 "was greater above the more pristine"

*Done*

Around line 25 I'm wondering about the monoterpene distribution & diel cycle at hyytiala

*To our knowledge, there were no speciated measurements of monoterpenes during BAECC. The study by Hakola et al., [2012 ACP] cited in the manuscript is the most recent work at the site.*

P. 12 line 9 I'm wondering how you assessed the role of boundary layer dynamics

*See above discussion in response to an earlier comment. Normalizing to the total OA focuses changes in relative composition not absolute abundance to avoid a direct effect of boundary layer height changes.*

Line 15: ... or difference bvoc mix making sources different, or different temperatures ... might end this is a little more open ended about explanation?

*With the added sentence near the beginning of this paragraph, we believe we have now conveyed that there are many differences between the two sites that need to be investigated further.*

P. 18 table 1: why is only gOC average mixing ratio reported in the caption?

*That was an example to show how ppt converts to ug m-3.*

Fig. 1 : why different units on panel b than elsewhere (ng m-3)? Do I interpret the righthand panels correctly to say that all dimer species are more abundant in the gas phase than particle? This seems surprising...

5 *Figure 1a shows gas-phase in units of ppt.*

*Figure 1b shows particle-phase in units of nmole m-3.*

Fig. 2 : are these all gas phase only data?

*We have revised the figure 2 caption to clarify that the data in (b) and (c) are both of*
10 *the gas-phase.*

Fig. 8: explain the "adjustment" a bit more – is this just no3 mass x 265/62?

*Yes. Figure 8 caption has been revised to show that more clearly.*

**Anonymous Referee #2**

5   Lee et al. describe aerosol and gas-phase measurements of organic compounds from tall tower located above a boreal forest. The measurements show the diurnal patterns of gas-phase species, measured using an I-CIMS, and particle-phase species, measured using a FIGAERO inlet. The authors find that most gas and particle-phase species exhibit either a morning, daytime, or nighttime enhancement. In the gas-

10 phase, smaller molecules dominated the organic distributions, though highly oxygenated molecules (or HOMs) were observed during the morning and daytime. In the particle phase, HOMs were observed in each diurnal subgroup. Of these compounds, the organic nitrates constituted a significant fraction of the detected organic species, with highest contributions at night. A non-negligible amount of

15 nitrate dimers were observed, which were suspected to be formed by the reaction between $NO_3RO_2$ + $RO_2$ radicals.

The results from this study contributes to the evidence that organic nitrate species formed from biogenic VOC oxidation significantly contribute to organic aerosol, especially at night. The results are interesting and well-interpreted, the paper is well

20 written, and the figures are nice and descriptive. I recommend the manuscript for publication provided that the authors address the following very minor comments.

*The authors greatly appreciate the reviewer for their detailed comments and suggestions.*

25   Page 4, lines 26 - It's not clear why NNMF was not applied to raw concentration counts. Is this to give equal weight to all species (i.e., the assumption is that changes in concentrations will be approximately equal across species)? Furthermore, how were the errors estimated? Please clarify.

*We perform NNMF on the deviation from the daily mean of each species, so*

*regardless of whether NNMF is performed using mixing ratio, mass concentration, or raw signal counts, each species is effectively treated with equal weight in this approach. Factorization will create groups that explain the largest fraction of total variance. The dynamic range of CIMS means that a few very large peaks will often dominate and mask other possible components. We have now clarified that on line 28, page 4. We do not account for uncertainties, though the precision error is negligible, and we restrict the approach to produce only a few of the dominant factors. There is likely a calibration uncertainty that is large, but difficult to quantify for individual species, hence, another motivation for giving all components an equal weight.*

Page, Lines 8 -12 - I really like this approach for resolving factors, especially as the authors are not trying to over-interpret the data. Can the authors mention how well the variability was explained by the resolved subgroups? Also, what type of residual was left over not explained by NNMF?

*We state that each species that was deemed as belonging to a given subgroup exhibited a correlation coefficient ($R2$) better than 0.45 with that subgroup's diel trend determined by NNMF (lines 7-9, page 5). The species that did not exhibit a sufficiently distinct enough diel trend, or the "others" subgroup, are effectively the residual.*

Page 5, Lines 21-22 - I'm confused by what the authors are trying to say here. Do the authors mean to say that high abundance masses observed in the gas phase were also observed in the particle phase, but that the presence of these species was unexpected based on volatility? Can the authors give some examples to help orient the reader? This would be useful when interpreting the results in Fig 1.

*We have now clarified that statement on page 5, lines 21-24. Organic compounds typically exhibit an approximate bell-shape distribution in the particle-phase, with the most abundant organic materials possessing molecular weight of ~220 g/mol. The exception appears to be the 50 or so species at the low molecular weight (~125) end that are nearly as abundant as the material with higher molecular weight. We assume*

*these compounds likely resulted from thermal fragmentation of higher molecular weight material (as described in Lopez-Hilfiker et al ACP 2015).*

Page 6, lines 1 - 3. Couldn't the variability also be explained, in part, due to higher emission rates of monoterpenes as a function of temperature?

*Yes, we have now clarified that comment on page 6, line 8.*

Page 7, lines 9-11. Do the authors have other data that could show whether the breakup of the nocturnal boundary layer contributed to the trends observed here? Were there vertically resolved measurements (e.g. temperature, RH, etc) that support the presented of a nocturnal layer below the tower? I realize that this will not change the interpretation of gas and particle phase correlations, but it would be interesting to know if the morning diel pattern is dominated by sudden burst of species produced during the night time, or by a sudden burst in oxygenated species once photochemistry kicked in.

*There are vertical profile temperature measurements from another tower at the same site that, along with published reports (i.e. Zha et al., 2018; Schobesberger et al., 2016), show that there is a de-coupling of air above and below the forest canopy at nighttime when the vertical mixing becomes relatively stagnant. However, without FIGAERO-CIMS measurements above and below the canopy, the influence of mixing on the observed diel trends is difficult to definitively conclude.*

Page 9, Lines 12-23. Is it reasonable to infer that the agreement between the AMS (located below the forest canopy) and FIGAERO CIMS (located above the forest canopy) in pON provides evidence that that the tall tower was within the nocturnal boundary layer?

*That is a keen observation by the reviewer. We strongly suspect there is strong nighttime decoupling of the air near the surface from above the canopy. That the diel*

*trends of AMS NO3 below the canopy and FIGAERO-CIMS pON above the canopy appear similar is likely due to the fact that organic nitrate production (due to NO3-radical-driven chemistry) and partitioning to the particle-phase (colder ambient temperature) are both relatively stronger at night compared to day.*

Figure 3: This figure is great and conveys a lot of information. Can the authors comment on what appears to be a bi-modal distribution in the C11-C20 compounds? There appears to be two peaks in the nO distributions, with one peaking around 5-6 oxygens, and the other peaking at 8-10 oxygens. Is this related to carbon number, or is
10   this explained more readily by other processes (auto-oxidation of dimers)?

*That is a great pickup on the part of the reviewer. We have added a statement on this on page 6, lines 27-33 noting this observation. There does appear to be a noticeable drop in abundance of C11 to C20 compounds (insets of figure 1b and 1d) that possess 7 oxygen atoms compared to those with 5-6 and 8-10. We speculate that such a*
15   *behavior is due to the combined effects of OH oxidation or ozonolysis and auto-oxidation leading to sequential addition of O2 that possibly do not favor the formation of O7 species, as well as volatilities of the resulting products that generally tends to decrease with increasing oxygen atom number. A detailed chemical model with observations from controlled laboratory experiments is needed to make a more*
20   *informed assessment.*

[revised manuscript text omitted]

Formatted ... [2]
Formatted ... [3]
Moved down [1]: A. N.
Moved (insertion) [1] ... [4]
Moved (insertion) [2] ... [5]
Moved down [2]: V. P.
Moved down [3]: R. J.
Moved (insertion) [3] ... [6]
Formatted ... [7]
Formatted ... [8]
Formatted ... [9]
Formatted ... [10]
Moved down [4]: H. G.
Moved (insertion) [4] ... [11]
Moved down [5]: P. O.
Moved (insertion) [5] ... [12]
... [13]
Formatted ... [14]
Formatted ... [15]
Formatted ... [16]
Formatted ... [17]
Formatted ... [18]
Formatted ... [19]
Formatted ... [20]
Formatted ... [21]
Formatted ... [22]
Formatted ... [23]
Formatted ... [24]
Formatted ... [25]
Formatted ... [26]
Formatted ... [27]
Formatted ... [28]
Moved down [6]: I. E.
Moved (insertion) [6] ... [29]
Formatted ... [30]
Formatted ... [31]
... [32]
Formatted ... [33]

[revised manuscript text omitted]

130 g mol⁻¹
71.8 ppt
n=602 | $C_{9.2}H_{14.9}N_{0.2}O_{3.2}$
181 g mol⁻¹
3.1 ppt
n=20 | $C_{7.5}H_{11.7}N_{0.5}O_{4.9}$
196 g mol⁻¹
8.9 ppt
n=92 | $C_{8.8}H_{14.4}N_{0.5}O_{4.9}$
204 g mol⁻¹
9.6 ppt
n=304 |
| gOC | $C_{4.5}H_{7.5}O_{3.7}$
121 g mol⁻¹
62 ppt
n=392 | $C_{9.1}H_{14.7}O_{2.8}$
169 g mol⁻¹
2.5 ppt
n=11 | $C_{7.2}H_{11.2}O_{4.2}$
182 g mol⁻¹
4.0 ppt
n=33 | $C_{8.5}H_{14.4}O_{4.1}$
181 g mol⁻¹
5.1 ppt
n=148 |
| gON | $C_{5.8}H_{9.8}NO_{5.7}$
184 g mol⁻¹
9.4 ppt
n=210 | $C_{9.6}H_{15.7}NO_{5.1}$
227 g mol⁻¹
0.6 ppt
n=9 | $C_{7.8}H_{12.1}NO_{5.6}$
209 g mol⁻¹
4.9 ppt
n=59 | $C_{9.3}H_{14.5}NO_{5.8}$
232 g mol⁻¹
4.5 ppt
n=156 |
| particle-phase | $C_{8.5}H_{12.8}N_{0.1}O_{5.8}$
212 g mol⁻¹
0.25 µg m⁻³
n=519 | $C_{12.7}H_{20.2}N_{0.9}O_{7.9}$
311 g mol⁻¹
0.031 µg m⁻³
n=125 | $C_{8.9}H_{13.5}N_{0.3}O_{5.7}$
216 g mol⁻¹
0.23 µg m⁻³
n=332 | $C_{9.4}H_{15.7}N_{0.1}O_{6.1}$
231 g mol⁻¹
4.0×10⁻³ µg m⁻³
n=42 |
| pOC | $C_{8.6}H_{13.1}O_{5.8}$
210 g mol⁻¹
0.23 µg m⁻³
n=378 | $C_{16.8}H_{29.2}O_{12.2}$
427 g mol⁻¹
3.0×10⁻³ µg m⁻³
n=28 | $C_{8.5}H_{13.0}O_{5.0}$
196 g mol⁻¹
0.16 µg m⁻³
n=151 | $C_{9.6}H_{15.9}O_{6.0}$
231 g mol⁻¹
3.0×10⁻³ µg m⁻³
n=27 |
| pON | $C_{7.4}H_{11.0}NO_{7.1}$
227 g mol⁻¹
0.022 µg m⁻³
n=141 | $C_{12.2}H_{19.0}NO_{7.3}$
297 g mol⁻¹
0.028 µg m⁻³
n=97 | $C_{9.9}H_{14.8}NO_{7.3}$
264 g mol⁻¹
0.067 µg m⁻³
n=181 | $C_{8.2}H_{14.2}NO_{6.3}$
228 g mol⁻¹
1.0×10⁻³ µg m⁻³
n=15 |

[revised manuscript text omitted]

**Anonymous Referee #1**

This paper presents a novel and interesting dataset on oxidized organic species contributing to both gas and aerosol phase organic aerosol in a remote boreal forest. The analysis is possible by use of a FIGAERO inlet to monitor gas and aerosol phase
5  separately but using the same I- mass spectrometer, and positive matrix factorization to sift the complex spectra into 3 primary factors with unique diel behaviors. The authors interpret their results as showing a strikingly (considering the remote location and low NOx) large contribution of particulate organic nitrate to the organic aerosol mass concentrations, especially at night. This is consistent with other recent work and
10  thus builds evidence for an increasing role for organonitrates in SOA production. This paper is likely to be of great interest to the SOA research community and I recommend publication following minor revisions.

*We thank the reviewer for their detailed comments and questions. They clearly reflect the time and attention paid to the review.*

General suggestions: 1) since you will ultimately compare the org nitrate contribution to results from Kiendler-Scharr et al around Europe, and SOAS, I suggest to include somewhere in your introduction the average NOx concentration and BVOC composition (is it exclusively a-pinene?) at Hyytiala. Then when you discuss the
20  surprisingly large nitrate contribution, you can point to the differences.

*Since the mixing ratios of monoterpenes vary greatly as a function of time of day just like NOx (which is shown in figure S1), we now cite two papers that include figures showing monoterpenes as a function of time in the section that discusses organic nitrates so interested readers can find the relevant figures and information (lines 8-9,*
25  *page 11). Based on these references, we estimate that in terms of OH and ozone reactivity, the two dominant monoterpenes are α-pinene and Δ-3-carene, where the latter is important but on a more episodic basis.*

2) in the methods discussion, it sounded like you only ran PMF on the gas phase data.

But I think you may have separately done both gas and aerosol? Or did you just use the same groupings as found by the gasphase PMF for both phases to make the later plots? Either way, please clarify in the text.

*Yes, NNMF was performed separately for each phase. We clarified the statement on lines 29-30, page 4. We have not run the factorization on the combined gas+particle phase timeseries. This could be of interest and will likely better fit into a forthcoming manuscript on factorization using the time-resolved thermogram information.*

3) Why does your analysis only include zero or one nitrogen per molecule? Were no molecules with two or more observed, or did you omit them from the analysis?

*There were 27 organic di-nitrates (nN=2) identified in the mass spectra recorded during the BAECC campaign. Though the presence of these di-nitrates are interesting in and of themselves, they represented a small mass fraction of the total organic aerosol mass as measured by the FIGAERO-CIMS. Additionally, there is often a fair amount of uncertainty in attribution of signal to di-nitrates because the mass spectral signals often overlap with more abundant non-nitrogen containing compounds. As such, they were omitted from this bulk analysis.*

4) The discussion of variability in figures 6 b and d serving as evidence for the short lifetimes of some species was confusing to me. I don't see significantly greater variability in those figures compared to e.g. daytime gON in panel a.

*Abundances of gaseous organic material* could *exhibit large diel variability given that they are byproducts of oxidants and BVOC, both of which exhibit large diel variabilities. Abundance of organic material in the particle-phase is governed by the integral of production and loss, where potentially the largest loss term is from physical removal of the particle, for example, due to wet deposition or horizontal transport, both on the order of days and lacking a diurnal pattern. We observed, as the reviewer also notes, a distinct diel variability in the contribution of each subgroup to the total OA (figure 6). This could only occur if the particulate organic material*

*also has a short (hours-long) lifetime or if there is a large production rate that is highly localized. Otherwise, their diel variability becomes dampened, or less distinct. We have now clarified the statements in the first paragraph of page 10 to reflect this discussion.*

5) what is the difference between positive matrix factorization and non-negative? Maybe add a line to the methods explaining the difference and why you chose the latter.

*We have now revised paragraph 3 of page 4. Briefly, NNMF and PMF are very*
10  *similar, but NNMF allows for "0" mass, and is part of the Matlab software package.*

6) can you account for the effect of boundary layer height changes, to help interpret the morning nitrate source?

*The boundary layer height was measured by a ceilometer during BAECC. However,*
15  *without FIGAERO-CIMS measurements above and below the boundary layer, the boundary layer height measurement alone does little to shed light on its effect on the observed diel trends. We also note that the boundary layer height (or cloud base height, which is what is measured by the ceilometer) does not routinely fall below the measurement height at the top of the tower. The same argument applies to the lack of*
20  *mixing between above and below the forest canopy at nighttime. Without knowing the distribution of organic compounds detected by the FIGAERO-CIMS below the canopy, it is difficult to comment on the extent of influence that vertical mixing had on the observed diel trends. We state now more clearly on lines 7-9 (page 10) that boundary layer dynamics can have an effect on the observed diel trends.*

7) it looks like there is higher pON during the hottest days of your study. Can you comment on this? Can temperature-dependent partitioning be ruled out in explaining any of the diel variation? (Also around p. 8 line 5)

*Figure 2 shows the mixing ratio of gaseous HOM monomers and dimers. We now clarify that in the figure caption. And yes, ambient temperature affects emission rates of monoterpenes (and likely soil NOx emissions), and often associated with stagnation (high pressure), therefore, it can be expected that absolute concentrations of organic*

5 *nitrates would increase with ambient temperature.*

8) P.6 around line 15 you state the yield must be less that 0.5 to explain decreasing Abundance with # of oxygens. Does this assume that whatever does not yield functionalization stays at the same O:C?

10 *Yes. We have now clarified that statement in the last paragraph of page 6.*

9) Does figure 6c mean there is no nighttime o3 chemistry? If all gas phase OC is in the daytime factor? Or is the nighttime factor actually just a nitrate factor and o3 chemistry would be grouped in the daytime OC factor even if there is some at night.

15 *The fact that much of the mass of non-nitrate gaseous organic material (shown on figure 6c) exhibits a daytime enhanced trend likely means that much of it is produced during daytime when BVOC emission rates are at their highest and that these oxidation products remain throughout the night when production has slowed. But, the above statement does not mean nighttime ozone chemistry is absent, only that in a*

20 *relative sense there is an enhancement in OC above the canopy due to daytime emissions and chemistry. Given that factorization will pick out groups with large relative variance, it is possible that the "nighttime factor" is dominated by a relatively larger nitrate variance. Ozone chemistry occurs day and night, and thus has a less pronounced local or diel variability, likely causing its contribution to be*

25 *split into multiple factors. These issues are problematic for interpretation of results from any factorization approach, as it is a statistical pattern not necessarily a causal pattern.*

10) figure 7: are the nighttime factors so much sparser MS because there's no autooxidation in there, since nitrates don't need that to be condensable enough?

*Slower rate of auto-oxidation due to lower ambient temperatures could be a factor why the FIGAERO-CIMS observed fewer organic species that belonged to the nighttime subgroup. But, see above for other possible effects of factorization artificially masking nighttime ozone chemistry. If the temperature is high enough, nighttime ozonolysis should lead to autoxidation. Determining the relative fates of RO2 (auto-oxidation, reaction with NO, RO2, or HO2) may be informative, and should be pursued in the future. However, nitrates are not necessarily more condensable, in fact, a nitrate with the same O/C as a non-nitrate (e.g. C10H15O8N C10H16O8), is expected (from group contribution estimates) to have a higher saturation vapor pressure, and thus be "less condensable".*

Minor technical edits: Abstract line 22: mention that this comparator site is in the SE US.

*Done*

Top of p. 3: suggest to remove the last line of the intro, so you end with the statement of what you add with this work.

*Done*

P.4line8: "asitisoforderafactorof2"

*Done*

Line 11: " and the interpretation of these observations"

*Done*

P. 5 line 7: "approach is that species exhibiting subtle differences...trends may be lumped into"

5   *Done*

Line 22: are you talking about levels greater than expected in the particle phase specifically? Clarify

*We have clarified the statement on page 5, line 27-28.*

P.6 line 29 " motivates the use of"

*Done*

P.7 Line 4 and elsewhere: "adhered to" sounds strange to me - how about belonged
15   to?

*Done*

Line 21: "imply that formation rates.... were sufficiently higher...during the day, consis- tent with modeling results specific to the SMEAR"

20   *Done*

Line 27: "accumulated in the nocturnal "

*Done*

P. 8 line 10: is the Yan study referenced at the same site & season? Or similar forest type? Suggest to add additional comment specifying, and then in the next lines clarify which study you mean when. "...summer of 2014 reported here observed most gaseous...whereas those previous measurements near the canopy floor.. summer of 2012 had observed "

*Done*

P. 9 line 4: I thought nN previously signified average number of and per molecule ? Different meaning here?

*The effective atom numbers are mass-weighted. If a subgroup is composed of only non-nitrates, the effective nN is 0. If all nitrates, then nN=1. If 50/50 by mass, then nN=0.5*

P.10 lines 30 and 32: ~0.35 and ~5%: make both fractions or both percents

*Done*

Line 31" However, in that study the pON"

*Done*

P.11 1 " BAECC were also consistent with other observations of unexpectedly high..."

*Done*

Line 6 "was greater above the more pristine"

*Done*

5  Around line 25 I'm wondering about the monoterpene distribution & diel cycle at hyytiala

*To our knowledge, there were no speciated measurements of monoterpenes during BAECC. The study by Hakola et al., [2012 ACP] cited in the manuscript is the most recent work at the site.*

P. 12 line 9 I'm wondering how you assessed the role of boundary layer dynamics

*See above discussion in response to an earlier comment. Normalizing to the total OA focuses changes in relative composition not absolute abundance to avoid a direct effect of boundary layer height changes.*

Line 15: ... or difference bvoc mix making sources different, or different temperatures ... might end this is a little more open ended about explanation?

*With the added sentence near the beginning of this paragraph, we believe we have now conveyed that there are many differences between the two sites that need to be*
20  *investigated further.*

P. 18 table 1: why is only gOC average mixing ratio reported in the caption?

*That was an example to show how ppt converts to ug m-3.*

Fig. 1 : why different units on panel b than elsewhere (ng m-3)? Do I interpret the righthand panels correctly to say that all dimer species are more abundant in the gas phase than particle? This seems surprising...

5 *Figure 1a shows gas-phase in units of ppt.*

*Figure 1b shows particle-phase in units of nmole m-3.*

Fig. 2 : are these all gas phase only data?

*We have revised the figure 2 caption to clarify that the data in (b) and (c) are both of*
10 *the gas-phase.*

Fig. 8: explain the "adjustment" a bit more – is this just no3 mass x 265/62?

*Yes. Figure 8 caption has been revised to show that more clearly.*

**Anonymous Referee #2**

20 Lee et al. describe aerosol and gas-phase measurements of organic compounds from tall tower located above a boreal forest. The measurements show the diurnal patterns of gas-phase species, measured using an I-CIMS, and particle-phase species, measured using a FIGAERO inlet. The authors find that most gas and particle-phase species exhibit either a morning, daytime, or nighttime enhancement. In the gasphase, smaller molecules dominated the organic distributions, though highly oxygenated molecules (or HOMs) were observed during the morning and daytime. In the particle phase, HOMs were observed in each diurnal subgroup. Of these compounds, the organic nitrates constituted a significant fraction of the detected organic species, with highest contributions at night. A non-negligible amount of nitrate dimers were observed, which were suspected to be formed by the reaction between NO3RO2 + RO2 radicals.

The results from this study contributes to the evidence that organic nitrate species formed from biogenic VOC oxidation significantly contribute to organic aerosol, especially at night. The results are interesting and well-interpreted, the paper is well written, and the figures are nice and descriptive. I recommend the manuscript for publication provided that the authors address the following very minor comments.

*The authors greatly appreciate the reviewer for their detailed comments and suggestions.*

Page 4, lines 26 - It's not clear why NNMF was not applied to raw concentration counts. Is this to give equal weight to all species (i.e., the assumption is that changes in concentrations will be approximately equal across species)? Furthermore, how were the errors estimated? Please clarify.

*We perform NNMF on the deviation from the daily mean of each species, so regardless of whether NNMF is performed using mixing ratio, mass concentration, or raw signal counts, each species is effectively treated with equal weight in this approach. Factorization will create groups that explain the largest fraction of total variance. The dynamic range of CIMS means that a few very large peaks will often dominate and mask other possible components. We have now clarified that on line 28, page 4. We do not account for uncertainties, though the precision error is negligible, and we restrict the approach to produce only a few of the dominant factors. There is likely a calibration uncertainty that is large, but difficult to quantify for individual species, hence, another motivation for giving all components an equal weight.*

Page, Lines 8 -12 - I really like this approach for resolving factors, especially as the authors are not trying to over-interpret the data. Can the authors mention how well the variability was explained by the resolved subgroups? Also, what type of residual was left over not explained by NNMF?

5 *We state that each species that was deemed as belonging to a given subgroup exhibited a correlation coefficient (R2) better than 0.45 with that subgroup's diel trend determined by NNMF (lines 7-9, page 5). The species that did not exhibit a sufficiently distinct enough diel trend, or the "others" subgroup, are effectively the residual.*

Page 5, Lines 21-22 - I'm confused by what the authors are trying to say here. Do the authors mean to say that high abundance masses observed in the gas phase were also observed in the particle phase, but that the presence of these species was unexpected based on volatility? Can the authors give some examples to help orient the reader?
15 This would be useful when interpreting the results in Fig 1.

*We have now clarified that statement on page 5, lines 21-24. Organic compounds typically exhibit an approximate bell-shape distribution in the particle-phase, with the most abundant organic materials possessing molecular weight of ~220 g/mol. The exception appears to be the 50 or so species at the low molecular weight (~125) end*
20 *that are nearly as abundant as the material with higher molecular weight. We assume these compounds likely resulted from thermal fragmentation of higher molecular weight material (as described in Lopez-Hilfiker et al ACP 2015).*

Page 6, lines 1 - 3. Couldn't the variability also be explained, in part, due to higher
25 emission rates of monoterpenes as a function of temperature?

*Yes, we have now clarified that comment on page 6, line 8.*

Page 7, lines 9-11. Do the authors have other data that could show whether the

breakup of the nocturnal boundary layer contributed to the trends observed here? Were there vertically resolved measurements (e.g. temperature, RH, etc) that support the presented of a nocturnal layer below the tower? I realize that this will not change the interpretation of gas and particle phase correlations, but it would be interesting to know if the morning diel pattern is dominated by sudden burst of species produced during the night time, or by a sudden burst in oxygenated species once photochemistry kicked in.

*There are vertical profile temperature measurements from another tower at the same site that, along with published reports (i.e. Zha et al., 2018; Schobesberger et al., 2016), show that there is a de-coupling of air above and below the forest canopy at nighttime when the vertical mixing becomes relatively stagnant. However, without FIGAERO-CIMS measurements above and below the canopy, the influence of mixing on the observed diel trends is difficult to definitively conclude.*

Page 9, Lines 12-23. Is it reasonable to infer that the agreement between the AMS (located below the forest canopy) and FIGAERO CIMS (located above the forest canopy) in pON provides evidence that that the tall tower was within the nocturnal boundary layer?

*That is a keen observation by the reviewer. We strongly suspect there is strong nighttime decoupling of the air near the surface from above the canopy. That the diel trends of AMS NO3 below the canopy and FIGAERO-CIMS pON above the canopy appear similar is likely due to the fact that organic nitrate production (due to NO3-radical-driven chemistry) and partitioning to the particle-phase (colder ambient temperature) are both relatively stronger at night compared to day.*

Figure 3: This figure is great and conveys a lot of information. Can the authors comment on what appears to be a bi-modal distribution in the C11-C20 compounds? There appears to be two peaks in the nO distributions, with one peaking around 5-6 oxygens, and the other peaking at 8-10 oxygens. Is this related to carbon number, or is this explained more readily by other processes (auto-oxidation of dimers)?

*That is a great pickup on the part of the reviewer. We have added a statement on this on page 6, lines 27-33 noting this observation. There does appear to be a noticeable drop in abundance of C11 to C20 compounds (insets of figure 1b and 1d) that possess 7 oxygen atoms compared to those with 5-6 and 8-10. We speculate that such a behavior is due to the combined effects of OH oxidation or ozonolysis and auto-oxidation leading to sequential addition of O2 that possibly do not favor the formation of O7 species, as well as volatilities of the resulting products that generally tends to decrease with increasing oxygen atom number. A detailed chemical model with observations from controlled laboratory experiments is needed to make a more informed assessment.*

| Page 13: [1] Formatted | Microsoft Office User | 7/13/18 8:34:00 AM |
|---|---|---|

Font:Times New Roman, 10 pt

| Page 13: [2] Formatted | Microsoft Office User | 7/13/18 8:35:00 AM |
|---|---|---|

Normal, Line spacing:  single

| Page 13: [3] Formatted | Microsoft Office User | 7/13/18 8:34:00 AM |
|---|---|---|

Subscript

| Page 13: [3] Formatted | Microsoft Office User | 7/13/18 8:34:00 AM |
|---|---|---|

Subscript

| Page 13: [3] Formatted | Microsoft Office User | 7/13/18 8:34:00 AM |
|---|---|---|

Subscript

| Page 13: [3] Formatted | Microsoft Office User | 7/13/18 8:34:00 AM |
|---|---|---|

Subscript

| Page 13: [3] Formatted | Microsoft Office User | 7/13/18 8:34:00 AM |
|---|---|---|

Subscript

| Page 13: [4] Moved from page 13 (Move #1) | Microsoft Office User | 7/13/18 7:21:00 AM |
|---|---|---|

A. N.

| Page 13: [5] Moved from page 13 (Move #2) | Microsoft Office User | 7/13/18 7:21:00 AM |
|---|---|---|

V. P.

| Page 13: [6] Moved from page 13 (Move #3) | Microsoft Office User | 7/13/18 7:21:00 AM |
|---|---|---|

R. J.

| Page 13: [7] Formatted | Microsoft Office User | 7/13/18 8:34:00 AM |
|---|---|---|

Font:Not Italic

| Page 13: [7] Formatted | Microsoft Office User | 7/13/18 8:34:00 AM |
|---|---|---|

Font:Not Italic

| Page 13: [8] Formatted | Microsoft Office User | 7/13/18 8:34:00 AM |
|---|---|---|

Font:Times New Roman, 10 pt

| Page 13: [8] Formatted | Microsoft Office User | 7/13/18 8:34:00 AM |
|---|---|---|

Font:Times New Roman, 10 pt

| Page 13: [9] Formatted | Microsoft Office User | 7/13/18 8:34:00 AM |
|---|---|---|

Font:Not Italic

| Page 13: [9] Formatted | Microsoft Office User | 7/13/18 8:34:00 AM |
|---|---|---|

Font:Not Italic

| Page 13: [9] Formatted | Microsoft Office User | 7/13/18 8:34:00 AM |
|---|---|---|

Font:Not Italic

| Page 13: [9] Formatted | Microsoft Office User | 7/13/18 8:34:00 AM |
|---|---|---|

Font:Not Italic

| Page 13: [10] Formatted | Microsoft Office User | 7/13/18 8:34:00 AM |
|---|---|---|

Font:Not Italic

| | | |
|---|---|---|
| **Page 13: [11] Moved from page 13 (Move #4)** | **Microsoft Office User** | **7/13/18 7:26:00 AM** |

H. G.

| | | |
|---|---|---|
| **Page 13: [12] Moved from page 13 (Move #5)** | **Microsoft Office User** | **7/13/18 7:26:00 AM** |

P. O.

| | | |
|---|---|---|
| **Page 13: [13] Deleted** | **Microsoft Office User** | **7/13/18 7:27:00 AM** |

(2011)

| | | |
|---|---|---|
| **Page 13: [13] Deleted** | **Microsoft Office User** | **7/13/18 7:27:00 AM** |

(2011)

| | | |
|---|---|---|
| **Page 13: [14] Formatted** | **Microsoft Office User** | **7/13/18 8:34:00 AM** |

Font:Not Italic

| | | |
|---|---|---|
| **Page 13: [14] Formatted** | **Microsoft Office User** | **7/13/18 8:34:00 AM** |

Font:Not Italic

| | | |
|---|---|---|
| **Page 13: [14] Formatted** | **Microsoft Office User** | **7/13/18 8:34:00 AM** |

Font:Not Italic

| | | |
|---|---|---|
| **Page 13: [14] Formatted** | **Microsoft Office User** | **7/13/18 8:34:00 AM** |

Font:Not Italic

| | | |
|---|---|---|
| **Page 13: [15] Formatted** | **Microsoft Office User** | **7/13/18 8:34:00 AM** |

Font:Times New Roman, 10 pt

| | | |
|---|---|---|
| **Page 13: [16] Formatted** | **Microsoft Office User** | **7/13/18 8:34:00 AM** |

Font:Not Italic

| | | |
|---|---|---|
| **Page 13: [16] Formatted** | **Microsoft Office User** | **7/13/18 8:34:00 AM** |

Font:Not Italic

| | | |
|---|---|---|
| **Page 13: [16] Formatted** | **Microsoft Office User** | **7/13/18 8:34:00 AM** |

Font:Not Italic

| | | |
|---|---|---|
| **Page 13: [16] Formatted** | **Microsoft Office User** | **7/13/18 8:34:00 AM** |

Font:Not Italic

| | | |
|---|---|---|
| **Page 13: [17] Formatted** | **Microsoft Office User** | **7/13/18 8:34:00 AM** |

Font:Times New Roman, 10 pt

| | | |
|---|---|---|
| **Page 13: [17] Formatted** | **Microsoft Office User** | **7/13/18 8:34:00 AM** |

Font:Times New Roman, 10 pt

| | | |
|---|---|---|
| **Page 13: [18] Formatted** | **Microsoft Office User** | **7/13/18 8:34:00 AM** |

Font:Not Italic

| | | |
|---|---|---|
| **Page 13: [18] Formatted** | **Microsoft Office User** | **7/13/18 8:34:00 AM** |

Font:Not Italic

| | | |
|---|---|---|
| **Page 13: [19] Formatted** | **Microsoft Office User** | **7/13/18 8:34:00 AM** |

Font:Times New Roman, 10 pt

| | | |
|---|---|---|
| **Page 13: [20] Formatted** | **Microsoft Office User** | **7/13/18 8:34:00 AM** |

Font:Not Italic

| Page 13: [20] Formatted | Microsoft Office User | 7/13/18 8:34:00 AM |
|---|---|---|

Font:Not Italic

| Page 13: [21] Formatted | Microsoft Office User | 7/13/18 8:34:00 AM |
|---|---|---|

Font:Times New Roman, 10 pt

| Page 13: [22] Formatted | Microsoft Office User | 7/13/18 8:34:00 AM |
|---|---|---|

Font:Not Italic

| Page 13: [22] Formatted | Microsoft Office User | 7/13/18 8:34:00 AM |
|---|---|---|

Font:Not Italic

| Page 13: [22] Formatted | Microsoft Office User | 7/13/18 8:34:00 AM |
|---|---|---|

Font:Not Italic

| Page 13: [23] Formatted | Microsoft Office User | 7/13/18 8:34:00 AM |
|---|---|---|

Font:Times New Roman, 10 pt

| Page 13: [24] Formatted | Microsoft Office User | 7/13/18 8:34:00 AM |
|---|---|---|

Font:Not Italic

| Page 13: [24] Formatted | Microsoft Office User | 7/13/18 8:34:00 AM |
|---|---|---|

Font:Not Italic

| Page 13: [24] Formatted | Microsoft Office User | 7/13/18 8:34:00 AM |
|---|---|---|

Font:Not Italic

| Page 13: [25] Formatted | Microsoft Office User | 7/13/18 8:34:00 AM |
|---|---|---|

Font:Times New Roman, 10 pt

| Page 13: [26] Formatted | Microsoft Office User | 7/13/18 8:34:00 AM |
|---|---|---|

Font:Not Italic

| Page 13: [26] Formatted | Microsoft Office User | 7/13/18 8:34:00 AM |
|---|---|---|

Font:Not Italic

| Page 13: [27] Formatted | Microsoft Office User | 7/13/18 8:34:00 AM |
|---|---|---|

Font:Times New Roman, 10 pt

| Page 13: [28] Formatted | Microsoft Office User | 7/13/18 8:34:00 AM |
|---|---|---|

Font:Not Italic

| Page 13: [28] Formatted | Microsoft Office User | 7/13/18 8:34:00 AM |
|---|---|---|

Font:Not Italic

| Page 13: [28] Formatted | Microsoft Office User | 7/13/18 8:34:00 AM |
|---|---|---|

Font:Not Italic

| Page 13: [28] Formatted | Microsoft Office User | 7/13/18 8:34:00 AM |
|---|---|---|

Font:Not Italic

| Page 13: [29] Moved from page 13 (Move #6) | Microsoft Office User | 7/13/18 7:58:00 AM |
|---|---|---|

I. E.

| Page 13: [30] Formatted | Microsoft Office User | 7/13/18 8:34:00 AM |
|---|---|---|

Font:Not Italic

| Page 13: [30] Formatted | Microsoft Office User | 7/13/18 8:34:00 AM |
|---|---|---|

Font:Not Italic

| Page 13: [30] Formatted | Microsoft Office User | 7/13/18 8:34:00 AM |
|---|---|---|

Font:Not Italic

| Page 13: [31] Formatted | Microsoft Office User | 7/13/18 8:34:00 AM |
|---|---|---|

Font:Times New Roman, 10 pt

| Page 13: [32] Deleted | Microsoft Office User | 7/13/18 7:58:00 AM |
|---|---|---|

Guenther, A., T. Karl, P. Harley, C. Wiedinmyer, P. I. Palmer, and C. Geron (2006),

| Page 13: [33] Formatted | Microsoft Office User | 7/13/18 8:34:00 AM |
|---|---|---|

Font:Not Italic

| Page 13: [33] Formatted | Microsoft Office User | 7/13/18 8:34:00 AM |
|---|---|---|

Font:Not Italic

| Page 13: [33] Formatted | Microsoft Office User | 7/13/18 8:34:00 AM |
|---|---|---|

Font:Not Italic

| Page 14: [34] Formatted | Microsoft Office User | 7/13/18 8:34:00 AM |
|---|---|---|

Font:Times New Roman, 10 pt

| Page 14: [34] Formatted | Microsoft Office User | 7/13/18 8:34:00 AM |
|---|---|---|

Font:Times New Roman, 10 pt

| Page 14: [35] Deleted | Microsoft Office User | 7/13/18 7:59:00 AM |
|---|---|---|

Guenther, A. B., X. Jiang, C. L. Heald, T. Sakulyanontvittaya, T. Duhl, L. K. Emmons, and X. Wang (2012),

| Page 14: [36] Formatted | Microsoft Office User | 7/13/18 8:34:00 AM |
|---|---|---|

Font:Not Italic

| Page 14: [36] Formatted | Microsoft Office User | 7/13/18 8:34:00 AM |
|---|---|---|

Font:Not Italic

| Page 14: [36] Formatted | Microsoft Office User | 7/13/18 8:34:00 AM |
|---|---|---|

Font:Not Italic

| Page 14: [36] Formatted | Microsoft Office User | 7/13/18 8:34:00 AM |
|---|---|---|

Font:Not Italic

| Page 14: [37] Formatted | Microsoft Office User | 7/13/18 8:34:00 AM |
|---|---|---|

Font:Times New Roman, 10 pt

| Page 14: [37] Formatted | Microsoft Office User | 7/13/18 8:34:00 AM |
|---|---|---|

Font:Times New Roman, 10 pt

| Page 14: [38] Deleted | Microsoft Office User | 7/13/18 7:59:00 AM |
|---|---|---|

Hakola, H., H. Hellen, M. Hemmila, J. Rinne, and M. Kulmala (2012),

| Page 14: [39] Formatted | Microsoft Office User | 7/13/18 8:34:00 AM |
|---|---|---|

Font:Not Italic

| Page 14: [39] Formatted | Microsoft Office User | 7/13/18 8:34:00 AM |
|---|---|---|

Font:Not Italic

| Page 14: [39] Formatted | Microsoft Office User | 7/13/18 8:34:00 AM |
|---|---|---|

Font:Not Italic

| Page 14: [39] Formatted | Microsoft Office User | 7/13/18 8:34:00 AM |
|---|---|---|

Font:Not Italic

| Page 14: [40] Formatted | Microsoft Office User | 7/13/18 8:34:00 AM |
|---|---|---|

Font:Times New Roman, 10 pt

| Page 14: [40] Formatted | Microsoft Office User | 7/13/18 8:34:00 AM |
|---|---|---|

Font:Times New Roman, 10 pt

| Page 14: [41] Formatted | Microsoft Office User | 7/13/18 8:34:00 AM |
|---|---|---|

Font:Not Italic

| Page 14: [41] Formatted | Microsoft Office User | 7/13/18 8:34:00 AM |
|---|---|---|

Font:Not Italic

| Page 14: [41] Formatted | Microsoft Office User | 7/13/18 8:34:00 AM |
|---|---|---|

Font:Not Italic

| Page 14: [42] Formatted | Microsoft Office User | 7/13/18 8:34:00 AM |
|---|---|---|

Font:Times New Roman, 10 pt

| Page 14: [42] Formatted | Microsoft Office User | 7/13/18 8:34:00 AM |
|---|---|---|

Font:Times New Roman, 10 pt

| Page 14: [43] Formatted | Microsoft Office User | 7/13/18 8:34:00 AM |
|---|---|---|

Font:Not Italic

| Page 14: [43] Formatted | Microsoft Office User | 7/13/18 8:34:00 AM |
|---|---|---|

Font:Not Italic

| Page 14: [44] Formatted | Microsoft Office User | 7/13/18 8:34:00 AM |
|---|---|---|

Font:Times New Roman, 10 pt

| Page 14: [44] Formatted | Microsoft Office User | 7/13/18 8:34:00 AM |
|---|---|---|

Font:Times New Roman, 10 pt

| Page 14: [45] Formatted | Microsoft Office User | 7/13/18 8:34:00 AM |
|---|---|---|

Font:Not Italic

| Page 14: [45] Formatted | Microsoft Office User | 7/13/18 8:34:00 AM |
|---|---|---|

Font:Not Italic

| Page 14: [45] Formatted | Microsoft Office User | 7/13/18 8:34:00 AM |
|---|---|---|

Font:Not Italic

| Page 14: [46] Formatted | Microsoft Office User | 7/13/18 8:34:00 AM |
|---|---|---|

Font:Not Italic

| Page 14: [47] Formatted | Microsoft Office User | 7/13/18 8:34:00 AM |
|---|---|---|

Font:Times New Roman, 10 pt

| Page 14: [47] Formatted | Microsoft Office User | 7/13/18 8:34:00 AM |
|---|---|---|

Font:Times New Roman, 10 pt

| Page 14: [48] Deleted | Microsoft Office User | 7/13/18 8:02:00 AM |
|---|---|---|

Holzinger, R., A. Lee, K. T. Paw, and A. H. Goldstein (2005),

| Page 14: [49] Formatted | Microsoft Office User | 7/13/18 8:34:00 AM |
|---|---|---|

Font:Not Italic

| Page 14: [49] Formatted | Microsoft Office User | 7/13/18 8:34:00 AM |
|---|---|---|

Font:Not Italic

| Page 14: [49] Formatted | Microsoft Office User | 7/13/18 8:34:00 AM |
|---|---|---|

Font:Not Italic

| Page 14: [49] Formatted | Microsoft Office User | 7/13/18 8:34:00 AM |
|---|---|---|

Font:Not Italic

| Page 14: [50] Formatted | Microsoft Office User | 7/13/18 8:34:00 AM |
|---|---|---|

Font:Times New Roman, 10 pt

| Page 14: [50] Formatted | Microsoft Office User | 7/13/18 8:34:00 AM |
|---|---|---|

Font:Times New Roman, 10 pt

| Page 14: [51] Deleted | Microsoft Office User | 7/13/18 8:02:00 AM |
|---|---|---|

Horii, C. V., J. W. Munger, S. C. Wofsy, M. Zahniser, D. Nelson, and J. B. McManus (2004),

| Page 14: [52] Formatted | Microsoft Office User | 7/13/18 8:34:00 AM |
|---|---|---|

Font:Not Italic

| Page 14: [52] Formatted | Microsoft Office User | 7/13/18 8:34:00 AM |
|---|---|---|

Font:Not Italic

| Page 14: [52] Formatted | Microsoft Office User | 7/13/18 8:34:00 AM |
|---|---|---|

Font:Not Italic

| Page 14: [53] Formatted | Microsoft Office User | 7/13/18 8:34:00 AM |
|---|---|---|

Font:Not Italic

| Page 14: [54] Formatted | Microsoft Office User | 7/13/18 8:34:00 AM |
|---|---|---|

Font:Times New Roman, 10 pt

| Page 14: [55] Formatted | Microsoft Office User | 7/13/18 8:34:00 AM |
|---|---|---|

Font:Not Italic

| Page 14: [55] Formatted | Microsoft Office User | 7/13/18 8:34:00 AM |
|---|---|---|

Font:Not Italic

| Page 14: [55] Formatted | Microsoft Office User | 7/13/18 8:34:00 AM |
|---|---|---|

Font:Not Italic

| Page 14: [56] Formatted | Microsoft Office User | 7/13/18 8:34:00 AM |
|---|---|---|

Font:Times New Roman, 10 pt

| Page 14: [56] Formatted | Microsoft Office User | 7/13/18 8:34:00 AM |
|---|---|---|

Font:Times New Roman, 10 pt

| Page 14: [57] Formatted | Microsoft Office User | 7/13/18 8:34:00 AM |
|---|---|---|

Font:Not Italic

| Page 14: [57] Formatted | Microsoft Office User | 7/13/18 8:34:00 AM |
|---|---|---|

Font:Not Italic

| Page 14: [57] Formatted | Microsoft Office User | 7/13/18 8:34:00 AM |
|---|---|---|

Font:Not Italic

| Page 14: [58] Formatted | Microsoft Office User | 7/13/18 8:34:00 AM |
|---|---|---|

Font:Times New Roman, 10 pt

| Page 14: [58] Formatted | Microsoft Office User | 7/13/18 8:34:00 AM |
|---|---|---|

Font:Times New Roman, 10 pt

| Page 14: [59] Formatted | Microsoft Office User | 7/13/18 8:34:00 AM |
|---|---|---|

Font:Not Italic

| Page 14: [59] Formatted | Microsoft Office User | 7/13/18 8:34:00 AM |
|---|---|---|

Font:Not Italic

| Page 14: [60] Formatted | Microsoft Office User | 7/13/18 8:34:00 AM |
|---|---|---|

Font:Times New Roman, 10 pt

| Page 14: [60] Formatted | Microsoft Office User | 7/13/18 8:34:00 AM |
|---|---|---|

Font:Times New Roman, 10 pt

| Page 14: [61] Formatted | Microsoft Office User | 7/13/18 8:34:00 AM |
|---|---|---|

Font:Not Italic

| Page 14: [61] Formatted | Microsoft Office User | 7/13/18 8:34:00 AM |
|---|---|---|

Font:Not Italic

| Page 14: [62] Formatted | Microsoft Office User | 7/13/18 8:34:00 AM |
|---|---|---|

Font:Times New Roman, 10 pt

| Page 14: [62] Formatted | Microsoft Office User | 7/13/18 8:34:00 AM |
|---|---|---|

Font:Times New Roman, 10 pt

| Page 14: [63] Formatted | Microsoft Office User | 7/13/18 8:34:00 AM |
|---|---|---|

Font:Not Italic

| Page 14: [63] Formatted | Microsoft Office User | 7/13/18 8:34:00 AM |
|---|---|---|

Font:Not Italic

| Page 14: [63] Formatted | Microsoft Office User | 7/13/18 8:34:00 AM |
|---|---|---|

Font:Not Italic

| Page 14: [64] Formatted | Microsoft Office User | 7/13/18 8:34:00 AM |
|---|---|---|

Font:Times New Roman, 10 pt

| Page 14: [64] Formatted | Microsoft Office User | 7/13/18 8:34:00 AM |
|---|---|---|

Font:Times New Roman, 10 pt

| Page 14: [65] Formatted | Microsoft Office User | 7/13/18 8:34:00 AM |
|---|---|---|

Font:Not Italic

| Page 14: [65] Formatted | Microsoft Office User | 7/13/18 8:34:00 AM |
|---|---|---|

Font:Not Italic

| Page 14: [65] Formatted | Microsoft Office User | 7/13/18 8:34:00 AM |
|---|---|---|

Font:Not Italic

| Page 14: [65] Formatted | Microsoft Office User | 7/13/18 8:34:00 AM |
|---|---|---|

Font:Not Italic

| Page 14: [66] Formatted | Microsoft Office User | 7/13/18 8:34:00 AM |
|---|---|---|

Font:Times New Roman, 10 pt

| Page 14: [66] Formatted | Microsoft Office User | 7/13/18 8:34:00 AM |
|---|---|---|

Font:Times New Roman, 10 pt

| Page 14: [67] Deleted | Microsoft Office User | 7/13/18 8:06:00 AM |
|---|---|---|

Kavouras, I. G., N. Mihalopoulos, and E. G. Stephanou (1998),

| Page 14: [68] Formatted | Microsoft Office User | 7/13/18 8:34:00 AM |
|---|---|---|

Font:Not Italic

| Page 14: [68] Formatted | Microsoft Office User | 7/13/18 8:34:00 AM |
|---|---|---|

Font:Not Italic

| Page 14: [69] Formatted | Microsoft Office User | 7/13/18 8:34:00 AM |
|---|---|---|

Font:Times New Roman, 10 pt

| Page 14: [69] Formatted | Microsoft Office User | 7/13/18 8:34:00 AM |
|---|---|---|

Font:Times New Roman, 10 pt

| Page 15: [70] Formatted | Microsoft Office User | 7/13/18 8:34:00 AM |
|---|---|---|

Font:Not Italic

| Page 15: [70] Formatted | Microsoft Office User | 7/13/18 8:34:00 AM |
|---|---|---|

Font:Not Italic

| Page 15: [70] Formatted | Microsoft Office User | 7/13/18 8:34:00 AM |
|---|---|---|

Font:Not Italic

| Page 15: [70] Formatted | Microsoft Office User | 7/13/18 8:34:00 AM |
|---|---|---|

Font:Not Italic

| Page 15: [71] Formatted | Microsoft Office User | 7/13/18 8:34:00 AM |
|---|---|---|

Font:Times New Roman, 10 pt

| Page 15: [71] Formatted | Microsoft Office User | 7/13/18 8:34:00 AM |
|---|---|---|

Font:Times New Roman, 10 pt

| Page 15: [72] Formatted | Microsoft Office User | 7/13/18 8:34:00 AM |
|---|---|---|

Font:Not Italic

| Page 15: [72] Formatted | Microsoft Office User | 7/13/18 8:34:00 AM |
|---|---|---|

Font:Not Italic

| Page 15: [72] Formatted | Microsoft Office User | 7/13/18 8:34:00 AM |
|---|---|---|

Font:Not Italic

| Page 15: [73] Deleted | Microsoft Office User | 7/13/18 8:13:00 AM |
|---|---|---|

Kulmala, M., et al. (

| Page 15: [73] Deleted | Microsoft Office User | 7/13/18 8:13:00 AM |
|---|---|---|

Kulmala, M., et al. (

| Page 15: [73] Deleted | Microsoft Office User | 7/13/18 8:13:00 AM |
|---|---|---|

Kulmala, M., et al. (

| Page 15: [73] Deleted | Microsoft Office User | 7/13/18 8:13:00 AM |
|---|---|---|

Kulmala, M., et al. (

| Page 15: [74] Formatted | Microsoft Office User | 7/13/18 8:34:00 AM |
|---|---|---|

Font:Not Italic

| Page 15: [74] Formatted | Microsoft Office User | 7/13/18 8:34:00 AM |
|---|---|---|

Font:Not Italic

| Page 15: [75] Formatted | Microsoft Office User | 7/13/18 8:34:00 AM |
|---|---|---|

Font:Times New Roman, 10 pt

| Page 15: [75] Formatted | Microsoft Office User | 7/13/18 8:34:00 AM |
|---|---|---|

Font:Times New Roman, 10 pt

| Page 15: [76] Formatted | Microsoft Office User | 7/13/18 8:34:00 AM |
|---|---|---|

Font:Not Italic

| Page 15: [76] Formatted | Microsoft Office User | 7/13/18 8:34:00 AM |
|---|---|---|

Font:Not Italic

| Page 15: [77] Formatted | Microsoft Office User | 7/13/18 8:34:00 AM |
|---|---|---|

Font:Times New Roman, 10 pt

| Page 15: [77] Formatted | Microsoft Office User | 7/13/18 8:34:00 AM |
|---|---|---|

Font:Times New Roman, 10 pt

| Page 15: [78] Formatted | Microsoft Office User | 7/13/18 8:34:00 AM |
|---|---|---|

Font:Not Italic

| Page 15: [78] Formatted | Microsoft Office User | 7/13/18 8:34:00 AM |
|---|---|---|

Font:Not Italic

| Page 15: [78] Formatted | Microsoft Office User | 7/13/18 8:34:00 AM |
|---|---|---|

Font:Not Italic

| Page 15: [78] Formatted | Microsoft Office User | 7/13/18 8:34:00 AM |
|---|---|---|

Font:Not Italic

| Page 15: [79] Formatted | Microsoft Office User | 7/13/18 8:34:00 AM |
|---|---|---|

Font:Times New Roman, 10 pt

| Page 15: [79] Formatted | Microsoft Office User | 7/13/18 8:34:00 AM |
|---|---|---|

Font:Times New Roman, 10 pt

| Page 15: [80] Formatted | Microsoft Office User | 7/13/18 8:34:00 AM |
|---|---|---|

Font:Not Italic

| Page 15: [80] Formatted | Microsoft Office User | 7/13/18 8:34:00 AM |
|---|---|---|

Font:Not Italic

| Page 15: [80] Formatted | Microsoft Office User | 7/13/18 8:34:00 AM |
|---|---|---|

Font:Not Italic

| Page 15: [81] Formatted | Microsoft Office User | 7/13/18 8:34:00 AM |
|---|---|---|

Font:Times New Roman, 10 pt

| Page 15: [81] Formatted | Microsoft Office User | 7/13/18 8:34:00 AM |
|---|---|---|

Font:Times New Roman, 10 pt

| Page 15: [82] Formatted | Microsoft Office User | 7/13/18 8:34:00 AM |
|---|---|---|

Font:Not Italic

| Page 15: [82] Formatted | Microsoft Office User | 7/13/18 8:34:00 AM |
|---|---|---|

Font:Not Italic

| Page 15: [82] Formatted | Microsoft Office User | 7/13/18 8:34:00 AM |
|---|---|---|

Font:Not Italic

| Page 15: [82] Formatted | Microsoft Office User | 7/13/18 8:34:00 AM |
|---|---|---|

Font:Not Italic

| Page 15: [82] Formatted | Microsoft Office User | 7/13/18 8:34:00 AM |
|---|---|---|

Font:Not Italic

| Page 15: [83] Formatted | Microsoft Office User | 7/13/18 8:34:00 AM |
|---|---|---|

Font:Times New Roman, 10 pt

| Page 15: [83] Formatted | Microsoft Office User | 7/13/18 8:34:00 AM |
|---|---|---|

Font:Times New Roman, 10 pt

| Page 15: [84] Formatted | Microsoft Office User | 7/13/18 8:34:00 AM |
|---|---|---|

Font:Not Italic

| Page 15: [84] Formatted | Microsoft Office User | 7/13/18 8:34:00 AM |
|---|---|---|

Font:Not Italic

| Page 15: [84] Formatted | Microsoft Office User | 7/13/18 8:34:00 AM |
|---|---|---|

Font:Not Italic

| Page 15: [84] Formatted | Microsoft Office User | 7/13/18 8:34:00 AM |
|---|---|---|

Font:Not Italic

| Page 15: [85] Formatted | Microsoft Office User | 7/13/18 8:34:00 AM |
|---|---|---|

Font:Times New Roman, 10 pt

| Page 15: [85] Formatted | Microsoft Office User | 7/13/18 8:34:00 AM |
|---|---|---|

Font:Times New Roman, 10 pt

| Page 15: [86] Formatted | Microsoft Office User | 7/13/18 8:34:00 AM |
|---|---|---|

Font:Not Italic

| Page 15: [86] Formatted | Microsoft Office User | 7/13/18 8:34:00 AM |
|---|---|---|

Font:Not Italic

| Page 15: [86] Formatted | Microsoft Office User | 7/13/18 8:34:00 AM |
|---|---|---|

Font:Not Italic

| Page 15: [87] Formatted | Microsoft Office User | 7/13/18 8:34:00 AM |
|---|---|---|

Font:Times New Roman, 10 pt

| Page 15: [87] Formatted | Microsoft Office User | 7/13/18 8:34:00 AM |
|---|---|---|

Font:Times New Roman, 10 pt

| Page 15: [88] Deleted | Microsoft Office User | 7/13/18 8:23:00 AM |
|---|---|---|

Lopez-Hilfiker, F. D., et al. (2016b),

| Page 15: [88] Deleted | Microsoft Office User | 7/13/18 8:23:00 AM |
|---|---|---|

Lopez-Hilfiker, F. D., et al. (2016b),

| Page 15: [89] Formatted | Microsoft Office User | 7/13/18 8:34:00 AM |
|---|---|---|

Font:Not Italic

| Page 15: [89] Formatted | Microsoft Office User | 7/13/18 8:34:00 AM |
|---|---|---|

Font:Not Italic

| Page 15: [89] Formatted | Microsoft Office User | 7/13/18 8:34:00 AM |
|---|---|---|

Font:Not Italic

| Page 15: [90] Formatted | Microsoft Office User | 7/13/18 8:34:00 AM |
|---|---|---|

Font:Times New Roman, 10 pt

| Page 15: [90] Formatted | Microsoft Office User | 7/13/18 8:34:00 AM |
|---|---|---|

Font:Times New Roman, 10 pt

| Page 15: [91] Formatted | Microsoft Office User | 7/13/18 8:34:00 AM |
|---|---|---|

Font:Not Italic

| Page 15: [91] Formatted | Microsoft Office User | 7/13/18 8:34:00 AM |
|---|---|---|

Font:Not Italic

| Page 15: [91] Formatted | Microsoft Office User | 7/13/18 8:34:00 AM |
|---|---|---|

Font:Not Italic

| Page 15: [91] Formatted | Microsoft Office User | 7/13/18 8:34:00 AM |
|---|---|---|

Font:Not Italic

| Page 15: [92] Formatted | Microsoft Office User | 7/13/18 8:34:00 AM |
|---|---|---|

Font:Times New Roman, 10 pt

| Page 15: [92] Formatted | Microsoft Office User | 7/13/18 8:34:00 AM |
|---|---|---|

Font:Times New Roman, 10 pt

| Page 16: [93] Formatted | Microsoft Office User | 7/13/18 8:34:00 AM |
|---|---|---|

Font:Not Italic

| Page 16: [93] Formatted | Microsoft Office User | 7/13/18 8:34:00 AM |
|---|---|---|

Font:Not Italic

| Page 16: [93] Formatted | Microsoft Office User | 7/13/18 8:34:00 AM |
|---|---|---|

Font:Not Italic

| Page 16: [93] Formatted | Microsoft Office User | 7/13/18 8:34:00 AM |
|---|---|---|

Font:Not Italic

| Page 16: [94] Formatted | Microsoft Office User | 7/13/18 8:34:00 AM |
|---|---|---|

Font:Times New Roman, 10 pt

| Page 16: [94] Formatted | Microsoft Office User | 7/13/18 8:34:00 AM |
|---|---|---|

Font:Times New Roman, 10 pt

| Page 16: [95] Formatted | Microsoft Office User | 7/13/18 8:34:00 AM |
|---|---|---|

Font:Not Italic

| Page 16: [95] Formatted | Microsoft Office User | 7/13/18 8:34:00 AM |
|---|---|---|

Font:Not Italic

| Page 16: [95] Formatted | Microsoft Office User | 7/13/18 8:34:00 AM |
|---|---|---|

Font:Not Italic

| Page 16: [95] Formatted | Microsoft Office User | 7/13/18 8:34:00 AM |
|---|---|---|

Font:Not Italic

| Page 16: [96] Formatted | Microsoft Office User | 7/13/18 8:34:00 AM |
|---|---|---|

Font:Times New Roman, 10 pt

| Page 16: [96] Formatted | Microsoft Office User | 7/13/18 8:34:00 AM |
|---|---|---|

Font:Times New Roman, 10 pt

| Page 16: [97] Formatted | Microsoft Office User | 7/13/18 8:34:00 AM |
|---|---|---|

Font:Not Italic

| Page 16: [97] Formatted | Microsoft Office User | 7/13/18 8:34:00 AM |
|---|---|---|

Font:Not Italic

| Page 16: [97] Formatted | Microsoft Office User | 7/13/18 8:34:00 AM |
|---|---|---|

Font:Not Italic

| Page 16: [97] Formatted | Microsoft Office User | 7/13/18 8:34:00 AM |
|---|---|---|

Font:Not Italic

| Page 16: [98] Formatted | Microsoft Office User | 7/13/18 8:34:00 AM |
|---|---|---|

Font:Times New Roman, 10 pt

| Page 16: [98] Formatted | Microsoft Office User | 7/13/18 8:34:00 AM |
|---|---|---|

Font:Times New Roman, 10 pt

| Page 16: [99] Formatted | Microsoft Office User | 7/13/18 8:34:00 AM |
|---|---|---|

Font:Not Italic

| Page 16: [99] Formatted | Microsoft Office User | 7/13/18 8:34:00 AM |
|---|---|---|

Font:Not Italic

| Page 16: [99] Formatted | Microsoft Office User | 7/13/18 8:34:00 AM |
|---|---|---|

Font:Not Italic

| Page 16: [100] Formatted | Microsoft Office User | 7/13/18 8:34:00 AM |
|---|---|---|

Font:Not Italic

| Page 16: [100] Formatted | Microsoft Office User | 7/13/18 8:34:00 AM |
|---|---|---|

Font:Not Italic

| Page 16: [101] Formatted | Microsoft Office User | 7/13/18 8:34:00 AM |
|---|---|---|

Font:Times New Roman, 10 pt

| Page 16: [101] Formatted | Microsoft Office User | 7/13/18 8:34:00 AM |
|---|---|---|

Font:Times New Roman, 10 pt

| Page 16: [102] Deleted | Microsoft Office User | 7/13/18 8:27:00 AM |
|---|---|---|

Petäjä, T., et al. (2016), Baecc

| Page 16: [102] Deleted | Microsoft Office User | 7/13/18 8:27:00 AM |
|---|---|---|

Petäjä, T., et al. (2016), Baecc

| Page 16: [103] Formatted | Microsoft Office User | 7/13/18 8:34:00 AM |
|---|---|---|

Font:Not Italic

| Page 16: [103] Formatted | Microsoft Office User | 7/13/18 8:34:00 AM |
|---|---|---|

Font:Not Italic

| Page 16: [103] Formatted | Microsoft Office User | 7/13/18 8:34:00 AM |
|---|---|---|

Font:Not Italic

| Page 16: [104] Formatted | Microsoft Office User | 7/13/18 8:34:00 AM |
|---|---|---|

Font:Times New Roman, 10 pt

| Page 16: [104] Formatted | Microsoft Office User | 7/13/18 8:34:00 AM |
|---|---|---|

Font:Times New Roman, 10 pt

| Page 16: [105] Formatted | Microsoft Office User | 7/13/18 8:34:00 AM |
|---|---|---|

Font:Not Italic

| Page 16: [105] Formatted | Microsoft Office User | 7/13/18 8:34:00 AM |
|---|---|---|

Font:Not Italic

| Page 16: [105] Formatted | Microsoft Office User | 7/13/18 8:34:00 AM |
|---|---|---|

Font:Not Italic

| Page 16: [106] Formatted | Microsoft Office User | 7/13/18 8:34:00 AM |
|---|---|---|

Font:Times New Roman, 10 pt

| Page 16: [106] Formatted | Microsoft Office User | 7/13/18 8:34:00 AM |
|---|---|---|

Font:Times New Roman, 10 pt

| Page 16: [107] Formatted | Microsoft Office User | 7/13/18 8:34:00 AM |
|---|---|---|

Font:Not Italic

| Page 16: [107] Formatted | Microsoft Office User | 7/13/18 8:34:00 AM |
|---|---|---|

Font:Not Italic

| Page 16: [107] Formatted | Microsoft Office User | 7/13/18 8:34:00 AM |
|---|---|---|

Font:Not Italic

| Page 16: [108] Formatted | Microsoft Office User | 7/13/18 8:34:00 AM |
|---|---|---|

Font:Times New Roman, 10 pt

| Page 16: [108] Formatted | Microsoft Office User | 7/13/18 8:34:00 AM |
|---|---|---|

Font:Times New Roman, 10 pt

| Page 16: [109] Formatted | Microsoft Office User | 7/13/18 8:34:00 AM |
|---|---|---|

Font:Times New Roman, 10 pt

| Page 16: [109] Formatted | Microsoft Office User | 7/13/18 8:34:00 AM |
|---|---|---|

Font:Times New Roman, 10 pt

| Page 16: [110] Formatted | Microsoft Office User | 7/13/18 8:34:00 AM |
|---|---|---|

Font:Not Italic

| Page 16: [110] Formatted | Microsoft Office User | 7/13/18 8:34:00 AM |
|---|---|---|

Font:Not Italic

| Page 16: [111] Deleted | Microsoft Office User | 7/13/18 8:31:00 AM |
|---|---|---|

A.

| Page 16: [111] Deleted | Microsoft Office User | 7/13/18 8:31:00 AM |
|---|---|---|

A.

| Page 16: [112] Formatted | Microsoft Office User | 7/13/18 8:34:00 AM |
|---|---|---|

Font:Not Italic

| Page 16: [112] Formatted | Microsoft Office User | 7/13/18 8:34:00 AM |
|---|---|---|

Font:Not Italic

| Page 16: [112] Formatted | Microsoft Office User | 7/13/18 8:34:00 AM |
|---|---|---|

Font:Not Italic

| Page 16: [113] Formatted | Microsoft Office User | 7/13/18 8:34:00 AM |
|---|---|---|

Font:Times New Roman, 10 pt

| Page 16: [113] Formatted | Microsoft Office User | 7/13/18 8:34:00 AM |
|---|---|---|

Font:Times New Roman, 10 pt

| Page 16: [114] Formatted | Microsoft Office User | 7/13/18 8:34:00 AM |
|---|---|---|

Font:Not Italic

| Page 16: [114] Formatted | Microsoft Office User | 7/13/18 8:34:00 AM |
|---|---|---|

Font:Not Italic

| Page 16: [115] Formatted | Microsoft Office User | 7/13/18 8:34:00 AM |
|---|---|---|

Font:Times New Roman, 10 pt

| Page 16: [115] Formatted | Microsoft Office User | 7/13/18 8:34:00 AM |
|---|---|---|

Font:Times New Roman, 10 pt

| Page 16: [116] Formatted | Microsoft Office User | 7/13/18 8:34:00 AM |
|---|---|---|

Font:Not Italic

| Page 16: [116] Formatted | Microsoft Office User | 7/13/18 8:34:00 AM |
|---|---|---|

Font:Not Italic

| Page 16: [116] Formatted | Microsoft Office User | 7/13/18 8:34:00 AM |
|---|---|---|

Font:Not Italic

| Page 16: [116] Formatted | Microsoft Office User | 7/13/18 8:34:00 AM |
|---|---|---|

Font:Not Italic